# Deep Legendre Transform

**Aleksey Minabutdinov**[*]
Center of Economic Research and RiskLab
ETH Zurich, Switzerland
aminabutdinov@ethz.ch

**Patrick Cheridito**
Department of Mathematics and RiskLab
ETH Zurich, Switzerland
patrickc@ethz.ch

## Abstract

We introduce a novel deep learning algorithm for computing convex conjugates of differentiable convex functions, a fundamental operation in convex analysis with various applications in different fields such as optimization, control theory, physics and economics. While traditional numerical methods suffer from the curse of dimensionality and become computationally intractable in high dimensions, more recent neural network-based approaches scale better, but have mostly been studied with the aim of solving optimal transport problems and require the solution of complicated optimization or max-min problems. Using an implicit Fenchel formulation of convex conjugation, our approach facilitates an efficient gradient-based framework for the minimization of approximation errors and, as a byproduct, also provides a posteriori estimates of the approximation accuracy. Numerical experiments demonstrate our method's ability to deliver accurate results across different high-dimensional examples. Moreover, by employing symbolic regression with Kolmogorov–Arnold networks, it is able to obtain the exact convex conjugates of specific convex functions. Code is available at https://github.com/lexmar07/Deep-Legendre-Transform

## 1 Introduction

In this paper, we introduce DLT, a deep learning method for computing convex conjugates, which play an important role in different fields such as convex analysis, optimization, control theory, physics, and economics.

The modern formulation of convex conjugation, also called Legendre–Fenchel transformation, was introduced by Fenchel [1949] [1] and defines the convex conjugate of a function $f: C \to \mathbb{R}$ on a nonempty subset $C \subseteq \mathbb{R}^d$ by

$$f^*(y) = \sup_{x \in C} \left\{ \langle x, y \rangle - f(x) \right\}, \quad y \in \mathbb{R}^d. \tag{1.1}$$

This definition does not need any special assumptions on the function $f$ or the set $C$ and makes sense for all $y \in \mathbb{R}^d$. Moreover, it directly follows from (1.1) that the conjugate function $f^*: \mathbb{R}^d \to (-\infty, \infty]$ is lower semicontinuous and convex.

---

[*]Corresponding author

[1]The Legendre transform—a fundamental operation for switching between dual formulations and variables—has broad utility across thermodynamics, mechanics, optimization, and economics. Key applications include: switching between thermodynamic potentials by exchanging extensive variables (entropy, volume) for intensive conjugates (temperature, pressure) in physics; constructing Moreau envelopes in variational analysis; deriving indirect utility and profit functions in economics; and computing convex potentials in optimal transport. Recent work has further demonstrated applications in approximate dynamic programming Sharifi Kolarijani and Mohajerin Esfahani [2023] and self-concordant smoothing Adeoye and Bemporad [2023]. This paper introduces the Deep Legendre Transform algorithm, establishes its theoretical foundations, and demonstrates its application to optimal-control problems governed by Hamilton–Jacobi equations.

39th Conference on Neural Information Processing Systems (NeurIPS 2025).

In the special case, where $f\colon C \to \mathbb{R}$ is a differentiable convex function on an open convex set $C \subseteq \mathbb{R}^d$, one has

$$f^*(y) = \langle (\nabla f)^{-1}(y), y \rangle - f((\nabla f)^{-1}(y)) \tag{1.2}$$

for all $y \in D = \nabla f(C)$. The right side of (1.2), called Legendre transform, was used by Legendre [1787] when studying the minimal surface problem. Note that for it to be well-defined, the gradient map $\nabla f\colon C \to D$ does not have to be invertible. In fact, it can easily be shown that if $f$ is differentiable and convex, the right side of (1.1) is maximized by any $x \in C$ satisfying $\nabla f(x) = y$. So the preimage $(\nabla f)^{-1}(y)$ in (1.2) can be substituted by any point $x \in (\nabla f)^{-1}(y)$.

Since $f^*$ is not always available in closed form, various numerical methods have been developed to approximate it numerically. Most of the classical approaches use the modern definition (1.1) and approximate $f^*(y)$ for $y$ in a finite subset $\mathcal{Y} \subseteq \mathbb{R}^d$ by the discrete Legendre–Fenchel transform

$$\max_{x \in \mathcal{X}} \{ \langle x, y \rangle - f(x) \} \tag{1.3}$$

based on a finite subset $\mathcal{X} \subseteq C$. Of course, for (1.3) to provide a satisfactory approximation to $f^*$, the subsets $\mathcal{X}$ and $\mathcal{Y}$ need to be sufficiently dense. Typically, $\mathcal{X}$ and $\mathcal{Y}$ are chosen as grids $\mathcal{X} = \mathcal{X}_1 \times \ldots \times \mathcal{X}_d$ and $\mathcal{Y} = \mathcal{Y}_1 \times \ldots \times \mathcal{Y}_d$ of similar size. For instance, for regular grids of the form $\mathcal{X} = \mathcal{Y} = I_N^d = \{ \frac{1}{N}, \frac{2}{N}, \ldots, 1 \}^d$, conditions under which the discrete Legendre–Fenchel transform (1.3) converges to $f^*$ have been given by Corrias [1996]. In this case, a naive computation of (1.3) for all $y \in \mathcal{Y}$, has complexity $O(N^{2d})$. For the case where $N$ is a power of $2$, Brenier [1989] and Noullez and Vergassola [1994] have derived fast algorithms to compute (1.3), similar in spirit to the fast Fourier transform of Cooley and Tukey [1965], with complexity $O((N \log N)^d)$. By first passing to convex hulls, Lucet [1997] has been able to design an algorithm with complexity $O(|\mathcal{X}| + |\mathcal{Y}|)$, which for $\mathcal{X} = \mathcal{Y} = I_N^d$ means $O(N^d)$. But since this is still exponential in $d$, it essentially limits the method to dimensions $d = 1, 2, 3$ and $4$.

A different approach, going back to a tweet by Peyré [2020], uses the fact that if $C \subseteq \mathbb{R}^d$ is bounded, $f^*$ can be approximated[2] with the softmax convex conjugate

$$f_\varepsilon^*(y) = \varepsilon \log \left( \int_C \exp \left( \frac{\langle x, y \rangle - f(x)}{\varepsilon} \right) dx \right)$$

for a small $\varepsilon > 0$, which can be expressed via Gaussian convolution as

$$f_\varepsilon^*(y) = Q_\varepsilon^{-1} \left( \frac{1}{Q_\varepsilon(f) * G_\varepsilon} \right)(y),$$

where

$$G_\varepsilon = \exp \left( -\frac{1}{2\varepsilon} \| \cdot \|_2^2 \right), \; Q_\varepsilon(f) = \exp \left( \frac{1}{2\varepsilon} \| \cdot \|_2^2 - \frac{1}{\varepsilon} f(\cdot) \right), \text{ and } Q_\varepsilon^{-1}(F) = \frac{1}{2} \| \cdot \|_2^2 - \varepsilon \log F(\cdot).$$

Approximating the convolution $Q_\varepsilon(f) * G_\varepsilon$ with a sum over a finite set $\mathcal{X} \subseteq C$ yields a GPU-friendly algorithm with complexity $O(|\mathcal{X}| \log |\mathcal{X}|)$; see Blondel and Roulet [2024] for details. But since it is grid-based, it still only gives accurate results in low-dimensional settings.

Lucet et al. [2009] have explored the computation of convex conjugates of bivariate piecewise linear quadratic convex functions, and Haque and Lucet [2018] have proposed an algorithm for such functions with linear worst-case complexity. But the scalability to higher dimensions remains a challenge.

Recently, numerical methods for convex conjugation have also been studied in the machine learning literature as a tool to solve the Kantorovich dual of optimal transport problems and develop efficient Wasserstein GANs. While their performance has mostly been tested to see how well they help to solve transport problems, promising results have been obtained by training neural networks to predict an optimal $x$ in the Legendre–Fenchel transform (1.1). E.g. Dam et al. [2019] and Taghvaei and Jalali [2019] predict $x$ from $y$ with a neural network $h_\vartheta(y)$, where $h_\vartheta$ is maximizing the right side of (1.1). Makkuva et al. [2020] do the same using predictors of the form $h_\vartheta = \nabla \ell_\vartheta$ for an ICNN[3]

---

[2] The entropic log-sum-exp smoothing of a max can be traced back to the Gibbs variational principle; brought into modern convex-optimisation analysis by Nesterov [2005].

[3] Input Convex Neural Network; see Amos et al. [2017]

$\ell_\vartheta$. Korotin et al. [2021a,b] and Amos [2023] have proposed to predict an optimal $x$ in (1.1) with a neural network $h_\vartheta$ by minimizing the first order condition

$$\sum_i \|\nabla f(h_\vartheta(y_i)) - y_i\|_2^2 \quad \text{for sample points } y_i \in D,$$

which, in effect, amounts to learning a generalized inverse of the gradient map $\nabla f \colon C \to D$; see Section 4 below.

In this paper, we employ the following implicit Legendre formulation of convex conjugation:

$$f^*(\nabla f(x)) = \langle x, \nabla f(x) \rangle - f(x), \quad x \in C, \tag{1.4}$$

for differentiable convex functions $f \colon C \to \mathbb{R}$ defined on an open convex set $C \subseteq \mathbb{R}^d$. This allows us to approximate $f^* \colon D = \nabla f(C) \to \mathbb{R}$ by training a machine learning model $g_\theta \colon D \to \mathbb{R}$ parametrized by a parameter $\theta$ from a set $\Theta$ such that

$$g_\theta(\nabla f(x)) \approx \langle x, \nabla f(x) \rangle - f(x) \quad \text{for all } x \in C.$$

DLT implements this by solving

$$\min_{\theta \in \Theta} \sum_{x \in \mathcal{X}_{\text{train}}} (g_\theta(\nabla f(x)) + f(x) - \langle x, \nabla f(x) \rangle)^2 \tag{1.5}$$

for a training set $\mathcal{X}_{\text{train}} \subseteq C$. This combines the following advantages:

(i) On the training points $x \in \mathcal{X}_{\text{train}}$, thanks to (1.4), the algorithm's target values $\langle x, \nabla f(x) \rangle - f(x)$ exactly agree with the true values $f^*(\nabla f(x))$.

(ii) If $g_\theta$ is chosen as a neural network, the approach readily scales to high-dimensional setups, and if combined with an ICNN as approximator, it is guaranteed to output a convex approximation to the true Legendre transform, which is known to be convex.

(iii) Specifying $g_\theta$ as a KAN[4] makes it possible to obtain exact solutions for certain convex functions.

(iv) The implicit Fenchel formulation (1.4) can also be used to derive a posteriori error estimates.

(i) is a theoretical advantage compared to existing approaches and, as illustrated in Section 5 below, yields good numerical results, in fact, limited only by the used NN architecture/computing power available. We demonstrate that DLT achieves state-of-the-art performance by comparing it to the direct learning of the convex conjugate (i.e., using $f^*$ as a target function in cases where it is available in closed form).

(ii) ICNNs have been introduced by Amos et al. [2017], and it has been shown by Chen et al. [2019] that they can approximate any convex function arbitrarily well on compacts. With neural networks as approximators, the minimization problem (1.5) can numerically be solved in high dimensions with a stochastic gradient descent algorithm. The choice of ICNNs ensures that the approximation is convex. This regularizes the learning problem and helps against overfitting. Moreover, it is helpful in various downstream tasks, such as optimization, control, or optimal transport, where convex conjugation can be utilized as a powerful tool to find optimal solutions. Alternatively, we use ResNet (He et al. [2016]), which in many multidimensional cases provides a better approximation than ICNN but does not guarantee convexity.

(iii) KANs have recently been introduced by Liu et al. [2025]. They are particularly powerful for approximation tasks requiring high precision. But in addition, they can also be used for symbolic regression to recover functions that have a closed form expressions in terms of special functions such as polynomials, $\exp$, $\log$, $\sin$ or $\cos$. In Section 5 below, we show that KANs can compute Legendre transforms of certain convex functions exactly.

(iv) As a byproduct, our approach yields a posteriori $L^2$-approximation guarantees. Indeed, by testing a candidate approximating function $g \colon C \to \mathbb{R}$ against the implicit Legendre transform (1.4) on a test set $\mathcal{X}_{\text{test}} \subseteq C$ sampled from a distribution $\mu$, it is possible to derive estimates of the $L^2$-approximation error $\|g - f^*\|_{L^2(D,\nu)}^2$ with respect to the push-forward $\nu = \mu \circ (\nabla f)^{-1}$ of $\mu$ under the gradient

---

[4]Kolmogorov–Arnold Network; see Liu et al. [2025]

map $\nabla f \colon C \to D$. This is valuable beyond the framework of this paper since it makes it possible to evaluate the performance of any numerical method for convex conjugation.

The training and test sets $\mathcal{X}_{\text{train}}, \mathcal{X}_{\text{test}} \subseteq C$ used for training and testing a candidate approximator $g_\theta$ of the convex conjugate $f^*$ can be generated by sampling from an arbitrary distribution $\mu$ on $C$. But the gradient map $\nabla f \colon C \to D$ is distorting $C$ and transforms any finite subset $\mathcal{X} \subseteq C$ into $\nabla f(\mathcal{X}) \subseteq D$. If one wants $\nabla f(\mathcal{X})$ to consist of independent realizations of a desired target distribution $\nu$ on $D = \nabla f(C)$, $\mathcal{X}$ has to be sampled from a distribution $\mu$ whose push-forward $\mu \circ (\nabla f)^{-1}$ is similar to $\nu$. We do this by training a neural network $h_\vartheta \colon D \to \mathbb{R}^d$ that approximates a generalized inverse of $\nabla f \colon C \to D$. Then target training and test sets $\mathcal{Y}_{\text{train}}, \mathcal{Y}_{\text{test}} \subseteq D$ can be generated according to a desired distribution $\nu$ on $D$, and the sets $\mathcal{X}_{\text{train}} = h_\vartheta(\mathcal{Y}_{\text{train}})$ and $\mathcal{X}_{\text{test}} = h_\vartheta(\mathcal{Y}_{\text{test}})$ can be used for training and testing $g_\theta$.

The remainder of the paper is organized as follows: In Section 2, we introduce DLT. In Section 3, we derive estimates of the $L^2$-approximation error of any candidate approximating function of $f^*$ with respect to a given probability measure $\nu$ on the image $D = \nabla f(C)$ of $C$ under the gradient map $\nabla f$. In Section 4, we learn a generalized inverse mapping of $\nabla f$ that improves the performance of DLT if $\nabla f$ is heavily distorting $C$ and is needed to estimate the $L^2$-approximation error with respect to a desired target distribution $\nu$ on $D$. In Section 5, we report the results of different numerical experiments. Sec 6 concludes. Proofs of theoretical results and additional numerical experiments are given in the Appendix.

## 2 Convex conjugation as a machine learning problem

Let $f \colon C \to \mathbb{R}$ be a differentiable convex function defined on an open convex subset $C \subseteq \mathbb{R}^d$. Our goal is to approximate the convex conjugate $f^* \colon \mathbb{R}^d \to (-\infty, \infty]$ on the image $D = \nabla f(C)$ of $C$ under the gradient map $\nabla f$.

By the Fenchel–Young inequality[5], one has

$$f(x) + f^*(y) \geq \langle x, y \rangle \quad \text{for all } x \in C \text{ and } y \in \mathbb{R}^d$$

with equality if and only if $y = \nabla f(x)$. As a result, one obtains that on $D$, the convex conjugate $f^*$ coincides with the Legendre transform

$$f^*(y) = \left\langle (\nabla f)^{-1}(y), y \right\rangle - f((\nabla f)^{-1}(y)), \quad y \in D,$$

where the preimage $(\nabla f)^{-1}(y)$ can be substituted by any $x \in C$ satisfying $\nabla f(x) = y$. However, while in many situations, the function $f$ and its gradient $\nabla f$ are readily accessible, the inverse gradient map $(\nabla f)^{-1}$ might not exist, or if it does, it is often not be available in closed form. Therefore, we use the implicit Legendre representation:

$$f^*(\nabla f(x)) = \langle x, \nabla f(x) \rangle - f(x), \quad x \in C, \tag{2.1}$$

which makes it possible to formulate the numerical approximation of $f^*$ as a machine learning problem. More precisely, for any machine learning model $g_\theta \colon D \to \mathbb{R}$ with parameter $\theta$ in a set $\Theta$, DLT approximates $f^*$ on $D \subseteq \mathbb{R}^d$ by minimizing the empirical squared error

$$\sum_{x \in \mathcal{X}_{\text{train}}} \left( g_\theta(\nabla f(x)) + f(x) - \langle x, \nabla f(x) \rangle \right)^2 \tag{2.2}$$

over $\theta \in \Theta$. In principle, $\mathcal{X}_{\text{train}}$ can be any finite set of training data points $x \in C$. But in Section 4 below, we describe how it can be assembled such that the image $\nabla f(\mathcal{X}_{\text{train}})$ is approximately distributed according to a given target distribution $\nu$ on $D = \nabla f(C)$.

Any numerical solution $\hat{\theta} \in \Theta$ of the minimization problem (2.2) yields an approximation $g_{\hat{\theta}} \colon D \to \mathbb{R}$ of the true conjugate function $f^* \colon D \to \mathbb{R}$. The question is how accurate the approximation is.

## 3 A posteriori approximation error estimates

Thanks to the implicit form of the Legendre transform (2.1) it is possible to estimate the $L^2$-approximation error of any candidate approximator $g \colon D \to \mathbb{R}$ of $f^* \colon D \to \mathbb{R}$ with a standard

---

[5]see e.g., Fenchel [1949] or Rockafellar [1970]

Monte Carlo average even if the true $f^*$ is not known. Let $\mathcal{X}_{\text{test}}$ be a test set of random points $x \in C$ independently sampled from a probability distribution $\mu$ over $C$. Then, due to (2.1), one has

$$\frac{1}{|\mathcal{X}_{\text{test}}|} \sum_{x \in \mathcal{X}_{\text{test}}} \left(g(\nabla f(x)) + f(x) - \langle x, \nabla f(x) \rangle\right)^2 = \frac{1}{|\mathcal{X}_{\text{test}}|} \sum_{x \in \mathcal{X}_{\text{test}}} \left(g(\nabla f(x)) - f^*(\nabla f(x))\right)^2,$$

which, by the law of large numbers, for $|\mathcal{X}_{\text{test}}| \to \infty$, converges to

$$\int_C \left(g(\nabla f(x)) - f^*(\nabla f(x))\right)^2 \mu(dx),$$

which, in turn, equals the squared $L^2$-approximation error

$$\int_D \left(g(y) - f^*(y)\right)^2 \nu(dy) = \|g - f^*\|_{L^2(D,\nu)}^2, \tag{3.1}$$

with respect to the push-forward $\nu = \mu \circ (\nabla f)^{-1}$ of the measure $\mu$. This result lets us report Monte Carlo estimates[6] of the error $\|g - f^*\|_{L^2(D,\nu)}^2$ as if $f^*$ were known in closed form.

## 4   Sampling from a given distribution in gradient space

The most straight-forward way to generate training and test sets $\mathcal{X}_{\text{train}}, \mathcal{X}_{\text{test}} \subseteq C$ is to sample independent random points according to a convenient distribution $\mu$ on $C$.

But if one wants to minimize the $L^2$-approximation error (3.1) with respect to a given distribution $\nu$ on $D = \nabla f(C)$, $\mu$ has to be chosen so that its push-forward $\mu \circ (\nabla f)^{-1}$ equals $\nu$. For instance, if the goal is a close approximation everywhere on $D$ for a bounded subset $D \subseteq \mathbb{R}^d$, e.g. a ball or hypercube in $\mathbb{R}^d$, one might want $\nu$ to be the uniform distribution on $D$. If $D$ is unbounded, e.g. $D = \mathbb{R}^d$, one might want $\nu$ to be a multivariate normal distribution. But if one is interested in a close approximation of $f^*$ in a neighborhood of a particular point $y_0 \in D$, it is better to choose a distribution $\nu$ that is concentrated around $y_0$.

To sample points $x$ in $C$ such that the distribution of $\nabla f(x)$ is close to $\nu$, we learn a generalized inverse of the gradient map $\nabla f \colon C \to D$. This can be done by training a neural network $h_\vartheta \colon D \to \mathbb{R}^d$ with parameter $\vartheta \in \mathbb{R}^q$ such that either

$$h_\vartheta \circ \nabla f(x) \approx x \quad \text{for } x \in C \quad \text{(4.1a)} \qquad \text{or} \qquad \nabla f \circ h_\vartheta(y) \approx y \quad \text{for } y \in D \quad \text{(4.1b)}$$

We choose a standard feed-forward neural network $h_\vartheta$ with two hidden layers and GELU-activation. To achieve (4.1a) or (4.1b), one can either minimize

$$\sum_{x \in \mathcal{W}_{\text{train}}} \left(h_\vartheta \circ \nabla f(x) - x\right)^2 \quad \text{(4.2a)} \qquad \text{or} \qquad \sum_{y \in \mathcal{Y}_{\text{train}}} \left(\nabla f \circ h_\vartheta(y) - y\right)^2 \quad \text{(4.2b)}$$

for suitable training sets $\mathcal{W}_{\text{train}} \subseteq C$ or $\mathcal{Y}_{\text{train}} \subseteq D$, respectively. (4.2a) and (4.2b) both have advantages and disadvantages. Therefore, we combine the two.

We first pretrain $h_\vartheta$ by minimizing (4.2a). This works even if the initialized network $h_\vartheta$ maps some points $y$ from $D$ to $\mathbb{R}^d \setminus C$. $\mathcal{W}_{\text{train}}$ can e.g. be sampled from a uniform distribution over $C$ if $C$ is bounded or a multivariate normal distribution if $C$ is unbounded. If $\nabla f$ is not invertible, (4.2a) can typically not be reduced to zero. But pretraining will force $h_\vartheta$ to map most of $D$ into $C$. After pretraining, we minimize a convex combination of (4.2a) and (4.2b) for $\mathcal{Y}_{\text{train}}$ independently sampled from $\nu$. In every training step, points $y \in \mathcal{Y}_{\text{train}}$ for which $h_\vartheta(y)$ is not in $C$ are omitted from training. The training procedure can be stopped if (4.2b) is sufficiently close to zero, yielding an approximate generalized inverse $h_{\hat{\vartheta}}$ of $\nabla f$. If $\mathcal{Y}_{\text{test}}$ is a test set independently sampled of $\mathcal{Y}_{\text{train}}$ according to $\nu$, then $\mathcal{X}_{\text{train}} = C \cap h_{\hat{\vartheta}}(\mathcal{Y}_{\text{train}})$ and $\mathcal{X}_{\text{test}} = C \cap h_{\hat{\vartheta}}(\mathcal{Y}_{\text{test}})$ can be used to train and test a given machine learning model $g_{\hat{\theta}}$ to approximate $f^*$.

---

[6]Confidence intervals of the estimator are provided in Lemma A.1 in Appendix A.

## 5 Numerical experiments

In this section, we report results of numerical approximations[7] of the convex conjugates of differentiable convex functions using DLT with MLPs, ResNets, ICNNs, and KANs. As an application, we use DLT to approximate solutions of Hamilton–Jacobi PDEs.

**Numerical approximations with MLPs, ResNets and ICNNs**

We first approximate $f^*\colon D \to \mathbb{R}$ with different neural networks such as standard multi-layer perceptrons (MLPs), residual networks (ResNets) and input convex neural networks (ICNNs). We use MLPs with two hidden layers containing 128 units and GELU activation. Our ResNets have two residual blocks, each containing two dense layers, resulting in a 4-layer architecture with skip connections and GELU activation. The employed MLP-ICNNs have the same architecture as the MLPs but constrain hidden weights to be non-negative and apply softplus activations to ensure convexity. The ICNNs are specified as in Amos et al. [2017], using softplus activation and non-negative weights in two hidden layers along with input skip connections. Parameters were initialized according to a Gaussian distribution, but for ICNN, the weights were squared to ensure non-negativity. For training, we used the Adam optimizer[8].

As a first test, we compare DLT in the three cases of Table 1, where $f^*$ is known analytically, to directly learning $f^*$ using pairs $(y_i, f^*(y_i))$ as targets.

Table 1: Functions $f$, domains, Legendre transforms, and dual images $D = \nabla f(C)$.

| Function | $f(x)$ | Domain $C$ | $f^*(y)$ | $D = \nabla f(C)$ |
|---|---|---|---|---|
| Quadratic | $\frac{1}{2}\sum x_i^2$ | $\mathcal{N}(0,1)^d$ | $\frac{1}{2}\sum y_i^2$ | $\mathcal{N}(0,1)^d$ |
| Neg-Log | $-\sum \log(x_i)$ | $\exp(U[-2.3, 2.3])^d$ | $-d - \sum \log(-y_i)$ | $\{-1/x : x \in C\} \approx [-10, -0.1]^d$ |
| Neg-Entropy | $\sum x_i \log x_i$ | $\exp(U[-2.3, 2.3])^d$ | $\sum \exp(y_i - 1)$ | $\{\log x + 1 : x \in C\} \approx [-1.3, 3.3]^d$ |

Table 2 shows that DLT matches the performance of direct learning both in terms of approximation accuracy and training time. This empirical finding is maintained across different network architectures and dimensions, confirming that the performance of DLT is only constrained by the neural networks' capacity to approximate multidimensional functions and not by its implicit formulation. In our experiments, ResNet provided the best approximations despite not guaranteeing convexity, while MLP-ICNN showed limitations due to the lack of skip connections compared to ICNN[9].

**Functions without closed form conjugate**  Most functions do not admit a closed form expression for the convex conjugate. However, DLT gives the same a posteriori error estimates as if $f^*$ were known explicitly and learned directly on $D$. For illustration, we consider the quadratic over linear function

$$f(x) = \frac{x_1^2 + \cdots + x_d^2 + 1}{x_1 + \cdots + x_d + 1}.$$

It is not defined everywhere on $\mathbb{R}^d$. But it can be shown that it is convex on the convex set $C = \{x \in \mathbb{R}^d : x_1 + \cdots + x_d > 0\}$; see e.g. Boyd and Vandenberghe [2004]. The $i$-th component of the gradient $\nabla f(x)$ is

$$\frac{x_i^2 + 2x_i + \sum_{j \neq i} x_j(2x_i - x_j) - 1}{(x_1 + \cdots + x_d + 1)^2}. \tag{5.1}$$

But the convex conjugate $f^*$ is not known in closed form, and the image $D = \nabla f(C)$ is a non-trivial subset of $\mathbb{R}^d$. Even though we do not know $f^*$ explicitly, DLT yields accurate ICNN approximations of $f^*$ on $D$. Here, we used an ICNN with 3 hidden layers and ELU activations, we sampled random

---

[7]All computations were performed on NVIDIA T4 GPU and Intel Core Xeon CPUs running Python 3.11.12. More experiments, including the algorithms by Lucet's and Peyré, are available in the supplementary material.

[8]see Kingma and Ba [2017]

[9]Additional multidimensional experiments confirming that DLT has the same performance as direct learning of known conjugates in terms of approximation accuracy and training time are given in the Appendix.

Table 2: Comparison of DLT with direct learning (Dir) for $d = 10$ and 50. Errors were computed with Monte Carlo on test set of 4096 i.i.d. random points sampled from $\text{Unif} \circ (\nabla f)^{-1}$.

| Function | Model | $d = 10$ | | $d = 50$ | |
|---|---|---|---|---|---|
| | | RMSE$_{\text{DLT}}$ / RMSE$_{\text{Dir}}$ | $t_{\text{train}}$ (s) | RMSE$_{\text{DLT}}$ / RMSE$_{\text{Dir}}$ | $t_{\text{train}}$ (s) |
| Quadratic | MLP | 2.96e-5/2.86e-5 | 86/84 | 3.55e-3/3.59e-3 | 88/87 |
| | MLP-ICNN | 6.21e+0/3.15e+1 | 87/77 | 6.46e+2/6.46e+2 | 87/77 |
| | ResNet | 2.43e-5/3.00e-5 | 107/101 | 2.92e-3/3.25e-3 | 124/120 |
| | ICNN | 1.79e-2/1.23e-2 | 93/93 | 7.18e-2/6.66e-2 | 97/98 |
| Neg. Log | MLP | 6.75e-4/8.43e-4 | 94/94 | 3.81e-1/2.81e-1 | 98/98 |
| | MLP-ICNN | 4.23e+0/4.23e+0 | 64/62 | 2.03e+1/2.03e+1 | 67/70 |
| | ResNet | 7.34e-4/9.22e-4 | 109/111 | 1.05e-1/1.23e-1 | 134/131 |
| | ICNN | 6.47e+0/4.22e+0 | 54/65 | 2.03e+1/2.06e+1 | 74/62 |
| Neg. Ent. | MLP | 1.06e-3/1.26e-3 | 96/100 | 2.71e-1/3.91e-1 | 100/101 |
| | MLP-ICNN | 4.56e+0/4.01e+0 | 93/101 | 7.83e+1/7.87e+1 | 98/105 |
| | ResNet | 6.62e-4/7.41e-4 | 110/116 | 6.93e-2/7.27e-2 | 133/135 |
| | ICNN | 1.31e-2/1.74e-2 | 99/107 | 9.39e+1/9.39e+1 | 60/69 |

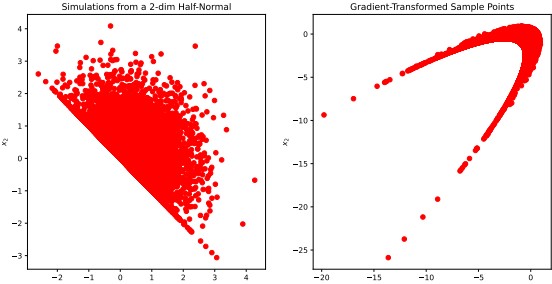

| $d$ | $\|g_\theta - f^*\|_{L^2}$ | Time (mm:ss) |
|---|---|---|
| 2 | 6.38e-04 | 4:14 |
| 5 | 2.39e-03 | 5:14 |
| 10 | 1.80e-02 | 6:55 |
| 20 | 3.67e-02 | 14:28 |
| 50 | 6.00e-02 | 29:19 |
| 100 | 3.68e-02 | 38:14 |

Figure 1: Quadratic-over-linear benchmark. **Left:** Sample points from a 2D standard half-normal distribution and their images under the gradient map $\nabla f$ in (5.1). **Right:** $L^2$ errors and training times for various dimensions.

points from a standard half-normal distribution on $C$. Figure 1 depicts these random points and their images under the gradient map $\nabla f$ given in (5.1) for $d = 2$. The table on the right shows estimates of the $L^2$-approximation error $\|g_{\hat{\theta}} - f^*\|_{L^2(\mathbb{R}^d, \nu)}$ together with the times in minutes and seconds it took to train the approximations $g_{\hat{\theta}}$ for different $d$. Note that the true $f^*$ is unknown here, but due to the results of Section 3, it is still possible to provide $L^2$ approximation errors.

In Appendix B, we study the following additional non-separable examples: (i) quadratic $f(x) = \frac{1}{2}x^\top Q x$; (ii) exponential-minus-linear $f(x) = e^{\langle a, x \rangle} - \langle b, x \rangle$; and (iii) ICNN-represented functions.

**Comparison with classical grid methods**

Next, we compare DLT against the LLT algorithm of Lucet [1997], a state-of-the-art grid-based method for computing convex conjugates. Rather than directly evaluating $\max_{x \in \mathcal{X}}\{\langle x, y \rangle - f(x)\}$ over a full $d$-dimensional grid $\mathcal{X}$ of size $N^d$ (which requires $O(N^{2d})$ operations), Lucet reformulates the $d$-dimensional maximization as a sequence of nested 1D problems:

$$f^*(y_1, \ldots, y_d) = \max_{x_1}\left\{ y_1 x_1 + \max_{x_2}\left\{ y_2 x_2 + \cdots + \max_{x_d}\left\{ y_d x_d - f(x) \right\} \right\} \right\}.$$

This decomposition reduces the computational complexity to $O(dN^{d+1})$ while still requiring $O(N^d)$ memory. Table 3 compares the methods for a fixed discretization of $N = 10$ points per dimension for two of the functions from Table 1. All errors were evaluated on the same random sample $\mathcal{Y}_{\text{test}} \subseteq D = \nabla f(C)$ against the true $f^*$. For low dimensions ($d \in \{2, 6\}$), Lucet runs quickly with modest memory, making it possible to use much finer grids within a reasonable time budget. However, as $d$ grows beyond 8, even modest grids become memory or time infeasible due to $O(N^d)$

Table 3: Comparison of Lucet with DLT for dimensions $d \in \{2, 6, 8, 10\}$. Lucet uses $N = 10$ points per dim; DLT uses a ResNet(128, 128). Reported: solution time $t_{\text{solve}}$, evaluation time $t_{\text{eval}}$, active memory (MB) and RMSE.

| Function | $d$ | Method | Model | $t_{\text{solve}}$ (s) | $t_{\text{eval}}$ (s) | MB(act) | RMSE |
|---|---|---|---|---|---|---|---|
| | 2 | Lucet | ($N^d$- Grid, $N{=}10$) | 0.00 | 0.12 | 0.0 | 3.65e−01 |
| | | DLT | (ResNet, 128,128) | 16.97 | 0.02 | 1.1 | 2.14e−02 |
| Neg. Log | 6 | Lucet | ($N^d$- Grid, $N{=}10$) | 15.17 | 1.02 | 15.3 | 1.83e+00 |
| | | DLT | (ResNet, 128,128) | 17.90 | 0.02 | 1.1 | 8.11e−02 |
| | 8 | Lucet | ($N^d$- Grid, $N{=}10$) | 1.96e+03 | 3.96 | 1.53e+03 | 2.93e+01 |
| | | DLT | (ResNet, 128,128) | 26.22 | 0.02 | 1.1 | 1.33e−01 |
| | 10 | DLT | (ResNet, 128,128) | 32.09 | 0.02 | 1.1 | 1.32e−01 |
| | 2 | Lucet | ($N^d$- Grid, $N{=}10$) | 0.00 | 0.12 | 0.0 | 1.42e−01 |
| | | DLT | (ResNet, 128,128) | 17.13 | 0.02 | 1.1 | 2.02e−02 |
| Neg. Entropy | 6 | Lucet | ($N^d$- Grid, $N{=}10$) | 15.11 | 1.01 | 15.3 | 7.32e+01 |
| | | DLT | (ResNet, 128,128) | 18.74 | 0.02 | 1.1 | 7.99e−02 |
| | 8 | Lucet | ($N^d$- Grid, $N{=}10$) | 1.94e+03 | 3.98 | 1.53e+03 | 1.08e+02 |
| | | DLT | (ResNet, 128,128) | 27.16 | 0.02 | 1.1 | 1.40e−01 |
| | 10 | DLT | (ResNet, 128,128) | 32.51 | 0.02 | 1.1 | 4.47e−01 |

storage and $O(d \cdot N^{d+1})$ computational complexity. In contrast, DLT maintains a constant memory footprint and scales smoothly to dimensions $d$ beyond 9.

**Direct vs. inverse sampling: distorting gradient mapping**

When the gradient mapping $\nabla f$ is significantly distorting, the image $\nabla f(\mathcal{X})$ of a uniformly sampled training set $\mathcal{X} \subseteq C$ will be highly non-uniformly distributed on $D$. This skewed distribution can adversely affect learning dynamics, concentrating approximation accuracy in certain regions while neglecting others. The approximate inverse sampling method introduced in Section 4 addresses this issue by allowing us to sample from any desired distribution (e.g., uniform) directly on $D$.

As an illustration, we consider the multidimensional negative logarithm $f(x) = -\sum_{i=1}^{d} \log(x_i)$ on $C = (10^{-3}, 10^{-1})^d$. With $\nabla f(x) = (-1/x_1, \ldots, -1/x_d)$, the gradient map severely distorts $C$, transforming it to $D = \nabla f(C) = (-1000, -10)^d$. The known closed-form of the convex conjugate is $f^*(y) = -\sum_{i=1}^{d} \log(-y_i) - d$.

First, we train with points sampled uniformly from $C$ (direct sampling). Then, we compare with our inverse sampling approach from Section 4, using $\mathcal{X}_{\text{train}} = h_{\hat{\vartheta}}(\mathcal{Y}_{\text{train}})$ where $\mathcal{Y}_{\text{train}}$ is sampled uniformly from $D$ and $h_{\hat{\vartheta}}$ approximates the inverse of $\nabla f$. Table 4 shows the $L^2$ approximation errors and training times for both methods.

The inverse sampling method consistently produces better approximations (1-2 orders of magnitude improvement for $d \geq 10$) at the cost of increased training time. Figure 2 illustrates this improvement for $d = 2$, where properly addressing the gradient distortion leads to a more uniform distribution of approximation accuracy across $D$.

**Applications to Hamilton–Jacobi equations**

The Hamilton–Jacobi (HJ) equation is a cornerstone of classical mechanics, optimal control theory, and mathematical physics, which provides a natural application domain for DLT.

Consider the Cauchy problem for the HJ equation:

$$\begin{cases} \partial_t u + H(\nabla_x u(x,t)) = 0, & (x,t) \in \mathbb{R}^d \times (0, \infty), \\ u(x,0) = g(x), & x \in \mathbb{R}^d, \end{cases} \tag{5.2}$$

for two convex functions $H, g : \mathbb{R}^d \to \mathbb{R}$.

The Hopf formula [Hopf, 1965] provides the following expression of $u$ as a double convex conjugate:

$$u(x,t) = (g^* + tH)^*(x), \tag{5.3}$$

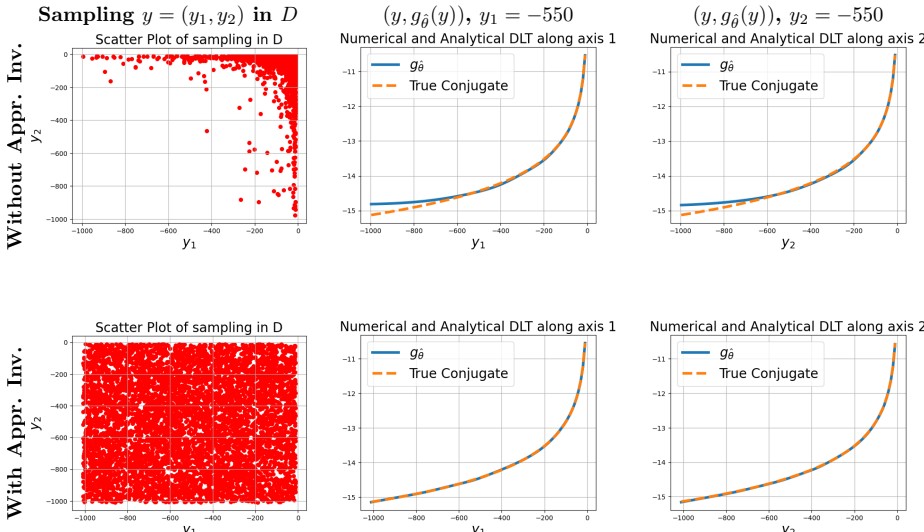

Figure 2: Negative Logarithm: Numerical estimate of the Legendre transform computed with and without inverse sampling compared to the true Legendre transform in two dimensions.

Table 4: Comparison of direct and inverse sampling for the negative logarithm. Inverse sampling uses an approximate inverse map $h_{\hat{v}}$ whose quality is measured by $2/(s\sqrt{d})\|\nabla f \circ h_{\hat{v}} - \mathrm{id}\|_{L^1}$, $s = 990$ is the side-length of $D$ and total training time split between $h_{\hat{v}}$ and $g_{\hat{\theta}}$.

| | Direct Sampling | | Inverse Sampling | | | | |
|---|---|---|---|---|---|---|---|
| $d$ | $\|g_{\hat{\theta}} - f^*\|_{L^2}$ | Time | Inv. Map Error | $\|g_{\hat{\theta}} - f^*\|_{L^2}$ | Time $h_{\hat{v}}$ | Time $g_{\hat{\theta}}$ | Total |
| 2 | 3.18e-02 | 12:50 | 1.32e-03 | 5.15e-03 | 4:06 | 22:15 | 26:21 |
| 5 | 2.47e-01 | 13:12 | 1.72e-03 | 3.45e-02 | 6:42 | 25:45 | 32:27 |
| 10 | 3.18e+00 | 14:41 | 2.17e-03 | 4.07e-02 | 9:16 | 31:22 | 40:38 |
| 20 | 4.31e+00 | 15:24 | 2.65e-03 | 7.72e-02 | 13:05 | 35:14 | 48:19 |
| 50 | 9.46e+00 | 17:51 | 7.98e-03 | 3.80e-01 | 23:31 | 44:24 | 67:55 |
| 100 | 5.44e+01 | 26:10 | 3.75e-02 | 3.06e+00 | 34:27 | 62:43 | 97:10 |

While this formula is explicit, evaluating it in high dimensions remains computationally challenging. To address this, we propose Time-DLT (Time-Parameterized Deep Legendre Transform) which approximates $(g^* + tH)^*$ using DLT with a neural network $u_\theta(x,t)$ with input variables $x$ and $t$.

**Experimental results.** We study an HJ equation (5.2) with $H(x) = g(x) = \frac{1}{2}\sum_{i=1}^{d} x_i^2$. It has the explicit solution $u(x,t) = \frac{\sum_{i=1}^{d} x_i^2}{2(1+t)}$, which we can use a reference. We use Time-DLT to compute a solution on $C = (-a,a)^d \times [0,T]$ for $a = T = 2$ and compare against the DGM (Deep Galerkin Method) of Sirignano and Spiliopoulos [2018], both implemented using the same ResNet architectures with two residual blocks. DGM minimizes the residual of the PDE on $C$ directly, while Time-DLT approximates the Hopf formula (5.3). Table 5 shows that Time-DLT outperforms DGM in approximating the true solution across different dimensions and times (however, we used the explicit expression for $g^*$ and just computed the outer Legendre transform).

Near $t = 0$, DGM performs better by directly enforcing the initial condition. But for larger $t$, the Time-DLT solutions are closer to the correct ones[10] indicated by a DGM/DLT $L^2$ error ratio $\rho > 1$. Note that DGM aims to minimize PDE residuals, but small PDE residuals do not always correspond to small approximation errors.

---

[10]We repeat the approximations 10 times to get statistically significant results and report estimated std. deviations $\sigma$.

Table 5: Comparison of DGM and Time-DLT for different dimensions and times.

| $d$ | $t$ | DGM | | | Time-DLT | | | Err. Ratio | |
|---|---|---|---|---|---|---|---|---|---|
| | | $L^2$ Error | PDE Res. | time (s) | $L^2$ Error | PDE Res. | time (s) | $\rho$ | $\sigma$ |
| 10 | 0.1 | 8.90e-2 | 4.82e-1 | 82.40 | 8.46e-2 | 1.60e+0 | 58.76 | 0.95 | 0.09 |
| | 0.5 | 2.14e-1 | 3.04e-1 | | 5.25e-2 | 9.17e-1 | | 3.69 | 0.44 |
| | 1.0 | 2.26e-1 | 1.85e-1 | | 3.68e-2 | 5.12e-1 | | 7.32 | 1.30 |
| | 2.0 | 1.64e-1 | 8.05e-2 | | 2.72e-2 | 2.27e-1 | | 8.64 | 2.57 |
| 30 | 0.1 | 3.45e-1 | 1.18e+0 | 87.18 | 4.29e-1 | 2.69e+0 | 192.62 | 0.83 | 0.12 |
| | 0.5 | 6.40e-1 | 8.85e-1 | | 2.60e-1 | 1.82e+0 | | 2.63 | 0.34 |
| | 1.0 | 1.01e+0 | 4.68e-1 | | 1.90e-1 | 9.69e-1 | | 5.15 | 1.23 |
| | 2.0 | 1.21e+0 | 2.56e-1 | | 1.28e-1 | 3.98e-1 | | 9.00 | 2.21 |

**Symbolic regression with KANs**

*Kolmogorov–Arnold networks* (KANs) have recently been introduced by Liu et al. [2025]. Inspired by the Kolmogorov–Arnold representation theorem, KANs hold great potential in approximation tasks requiring high precision. Moreover, in cases, where the Legendre transform has a closed form expression in terms of special functions, KANs can be employed to obtain the exact solution using symbolic regression. This is a significant advantage compared to numerical approximation since it uncovers the underlying mathematical structure. To illustrate symbolic regression with KANs, we consider the three functions of Table 1 in two dimensions.

In all three cases we performed a symbolic regression with KANs to minimize the empirical squared error (2.2) for a training set $\mathcal{X}_{\mathrm{train}}$ sampled uniformly from the square $[1, 10]^2$. We used polynomials up to the third order, $\exp$ and $\log$ as basis functions. As can be seen from the results in Table 6, KANs are able to find the correct symbolic expressions for the Legendre transforms of these functions with the right constants (up to rounding errors).

Table 6: DLT with KANs: two-dimensional training regions, $L^2$-errors and obtained symbolic expressions.

| Function | $C$ | $D$ | $L^2$ Error | Symbolic Expression |
|---|---|---|---|---|
| $\sum_{i=1}^{2} x_i^2$ | $[1, 10]^2$ | $[2, 20]^2$ | 0 | $0.25y_1^2 + 0.25y_2^2$ |
| $-\sum_{i=1}^{2} \log(x_i)$ | $[1, 10]^2$ | $[-1, -0.1]^2$ | 5.42e-8 | $-1.0\log(-0.70y_1) - 1.0\log(-6.09y_2) - 0.56$ |
| $\sum_{i=1}^{2} x_i \log(x_i)$ | $[1, 10]^2$ | $[1, 3.3]^2$ | 6.78e-05 | $0.37\exp(1.0y_1) + 0.37\exp(1.0y_2)$ |

# 6   Conclusion, limitation and future work

In this paper, we have introduced DLT (Deep Legendre Transform), a deep learning method for computing convex conjugates of differentiable convex functions $f\colon C \to \mathbb{R}$ defined on an open convex set $C \subseteq \mathbb{R}^d$. Our approach provides a posteriori approximation error estimates and scales effectively to high-dimensional settings, outperforming traditional numerical methods.

The *limitation* of DLT is that it needs a function $f\colon C \to \mathbb{R}$ to be differentiable and convex. Moreover, it only approximates the convex conjugate $f^*$, which can be defined on all of $\mathbb{R}^d$, on the image $D = \nabla f(C)$, which is contained in, but in general not equal to, $\mathrm{dom}\, f^* = \left\{y \in \mathbb{R}^d : f^*(y) < \infty\right\}$.

The extension of the approach to non-differentiable convex functions $f\colon C \to \mathbb{R}$ and approximation of $f^*$ on $\mathrm{dom}\, f^*$ will be studied in future work.

# Acknowledgments and Disclosure of Funding

This work was supported by the Swiss National Science Foundation under Grant No. 10003723. We are grateful to the anonymous reviewers for their constructive comments and suggestions.

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

# A  Unbiased a posteriori error estimator

**Lemma A.1** (**Unbiased a posteriori error estimator**). *Let $f\colon C \to \mathbb{R}$ be a differentiable convex function on an open convex set $C \subseteq \mathbb{R}^d$ and denote $D = \nabla f(C)$. Fix a probability measure $\mu$ on $C$ and consider the push-forward $\nu = \mu \circ (\nabla f)^{-1}$ on $D$. For a measurable function $g\colon D \to \mathbb{R}$, denote*

$$\widehat{\mathcal{E}}_n(g) = \frac{1}{n} \sum_{i=1}^{n} \Big( g(\nabla f(X_i)) + f(X_i) - \langle X_i, \nabla f(X_i) \rangle \Big)^2,$$

*where $X_1, \ldots, X_n$ are i.i.d. random vectors with distribution $\mu$. Then the following hold:*

(i) *If $g - f^* \in L^2(D, \nu)$, one has*

$$\mathbb{E}\Big[\widehat{\mathcal{E}}_n(g)\Big] = \int_D \Big( g(y) - f^*(y) \Big)^2 \nu(dy) = \|g - f^*\|_{L^2(D,\nu)}^2 \quad \text{(Unbiasedness)}$$

*and $\widehat{\mathcal{E}}_n(g) \to \|g - f^*\|_{L^2(D,\nu)}^2$ almost surely for $n \to \infty$.*

(ii) *If $g - f^* \in L^4(D, \nu)$, then*

$$\mathrm{Var}\Big(\widehat{\mathcal{E}}_n(g)\Big) = \frac{\sigma^2}{n} \quad \text{for} \quad \sigma^2 = \mathrm{Var}\Big( (g(\nabla f(X_1)) + f(X_1) - \langle X_1, \nabla f(X_1) \rangle)^2 \Big)$$

*and $\sqrt{n}\Big(\widehat{\mathcal{E}}_n(g) - \|g - f^*\|_{L^2(D,\nu)}^2\Big) \to \mathcal{N}(0, \sigma^2)$ in distribution as $n \to \infty$. In particular, if $\widehat{\sigma}$ is an estimate of $\sigma$ and $z_\alpha$ denotes the $\alpha$-quantile of $\mathcal{N}(0,1)$, then*

$$\widehat{\mathcal{E}}_n(g) \pm \frac{\widehat{\sigma}}{\sqrt{n}} z_{1-\alpha/2}$$

*is an approximate $(1 - \alpha)$-confidence interval for $\|g - f^*\|_{L^2(D,\nu)}^2$.*

*Proof.* (i) By the Fenchel–Young equality $f^*(\nabla f(x)) = \langle x, \nabla f(x) \rangle - f(x)$, one has

$$\Big( g(\nabla f(X_i)) + f(X_i) - \langle X_i, \nabla f(X_i) \rangle \Big)^2 = (g(\nabla f(X_i)) - f^*(\nabla f(X_i)))^2.$$

So, taking expectation and using the push-forward measure $\nu = \mu \circ (\nabla f)^{-1}$ yields

$$\mathbb{E}\Big[\widehat{\mathcal{E}}_n(g)\Big] = \int_C (g(\nabla f(x)) - f^*(\nabla f(x)))^2 \, \mu(dx) = \int_D (g(y) - f^*(y))^2 \, \nu(dy) = \|g - f^*\|_{L^2(D,\nu)}^2,$$

and it follows from the law of large numbers that $\widehat{\mathcal{E}}_n(g) \to \|g - f^*\|_{L^2(D,\nu)}^2$ almost surely. This shows (i).

(ii) Denote

$$Z_i = \Big( g(\nabla f(X_i)) + f(X_i) - \langle X_i, \nabla f(X_i) \rangle \Big)^2.$$

Then $\widehat{\mathcal{E}}_n(g) = \frac{1}{n} \sum_{i=1}^{n} Z_i$, and therefore,

$$\mathrm{Var}\Big(\widehat{\mathcal{E}}_n(g)\Big) = \frac{\mathrm{Var}(Z_1)}{n} = \frac{\sigma^2}{n}.$$

So, it follows from the central limit theorem that $\sqrt{n}\Big(\widehat{\mathcal{E}}_n(g) - \|g - f^*\|_{L^2(D,\nu)}^2\Big) \to \mathcal{N}(0, \sigma^2)$ in distribution. In particular, $\widehat{\mathcal{E}}_n(g) \pm \frac{\widehat{\sigma}}{\sqrt{n}} z_{1-\alpha/2}$ is an approximate $(1 - \alpha)$-confidence interval for $\|g - f^*\|_{L^2(D,\nu)}^2$, which concludes the proof. $\square$

# B  Technical appendix: additional experiments

**Content**

Sections B.1 and B.2 describe low-dimensional experiments with grid-based methods: Lucet's Linear Time Legendre Transform and Peyré's Gaussian convolution approach with Monte Carlo integration.

Section B.3 presents comprehensive benchmarks comparing our DLT approach against direct supervised learning when analytical conjugates are known. Using identical neural architectures across multiple functions and dimensions $d \in \{2, \ldots, 200\}$, we demonstrate that DLT achieves equivalent $L^2$ errors without requiring closed-form Legendre transforms.

Section B.4 evaluates DLT on challenging non-separable convex functions. We test quadratic forms with random matrices, exponential-minus-linear functions, pretrained ICNNs, and coupled softplus functions with quadratic pairwise interactions. These experiments show that DLT maintains accuracy across strongly coupled functions, although training time increases due to the more complex computation tree required for gradient evaluation in non-separable functions.

Section B.5 compares DLT with an alternative algorithm that uses an approximate inverse gradient mapping instead of exact gradients for training. While DLT trains on exact targets using the Fenchel–Young identity with true gradient values, the alternative method must learn an approximate inverse mapping to construct its training targets. We show that although such an approximation is feasible, it fundamentally depends on the quality of the inverse gradient approximation and provides weaker theoretical guarantees.

Section B.6 describes DLT applications to Hamilton–Jacobi PDEs via the Hopf formula. We introduce Time-Parameterized DLT that approximates $(g^* + tH)^*$ and compare against Deep Galerkin Methods, showing superior accuracy.

Section B.7 provides additional experiments with approximate inverse sampling for targeting specific dual space regions.

Section B.8 explores KANs integration with DLT for symbolic regression tasks, investigating whether approximate inverse sampling improves symbolic expression recovery in low-dimensional settings.

**Grid-based methods: definition-based method and Linear Time Legendre Transform**

**The definition-based method**

The definition-based method implements convex conjugation directly according to the Legendre–Fenchel definition (1.1)

$$f^*(s) = \sup_{x \in X} \{\langle s, x \rangle - f(x)\}$$

In explicit multi-dimensional form:

$$f^*(s_1, s_2, \ldots, s_d) = \max_{x_1, x_2, \ldots, x_d} \{s_1 x_1 + s_2 x_2 + \cdots + s_d x_d - f(x_1, x_2, \ldots, x_d)\}$$

**Key characteristics of the definition based method**

1. **Brute-force approach**: The algorithm exhaustively evaluates all possible coordinate combinations for each slope combination.

2. **Computational complexity**: $\mathcal{O}(N^{2d})$, where $N$ is the number of grid points per dimension and $d$ is the dimensionality. This complexity arises from:
   - $\mathcal{O}(N^d)$ operations to iterate through all slope combinations
   - $\mathcal{O}(N^d)$ operations to search through all coordinate combinations for each slope

3. **Memory usage**: $\mathcal{O}(N^d)$ for storing the output array, with minimal additional temporary storage.

The definition based method provides a baseline implementation before considering more contemporary approaches like the Linear-time Legendre Transform (LLT).

**The nested formulation**

The nested method exploits the mathematical structure of the Legendre–Fenchel transform by reformulating it as a sequence of nested maximization problems. For a function $f(x)$ in $d$ dimensions, the transform can be rewritten in nested form

$$f^*(s_1, s_2, \ldots, s_d) = \max_{x_1} \left\{ s_1 x_1 + \max_{x_2} \left\{ s_2 x_2 + \cdots + \max_{x_d} \left\{ s_d x_d - f(x_1, x_2, \ldots, x_d) \right\} \right\} \right\}$$

This decomposition allows dimension-by-dimension computation, starting from the innermost term.

**Algorithm structure**

**1D Linear-time Legendre Transform (LLT)**   The LLT algorithm by Lucet [1997] efficiently computes $f^*(s)$ in $\mathcal{O}(n + m)$ time, where $n$ is the number of data points and $m$ is the number of slopes:

1. **Convex hull step**: Compute the convex hull of the discrete data points $(x_i, f(x_i))$.

2. **Slope computation**: Calculate slopes between consecutive vertices of the convex hull:
$$c_i = \frac{f(x_{i+1}) - f(x_i)}{x_{i+1} - x_i}$$

3. **Merging step**: For each target slope $s_j$, find the interval $[c_i, c_{i+1}]$ containing $s_j$, then compute:
$$f^*(s_j) = s_j \cdot x_i - f(x_i)$$

$d$**-dimensional algorithm**   The nested approach proceeds in $d$ stages:

1. Compute the innermost term:
$$V_d(x_1, \ldots, x_{d-1}, s_d) = \max_{x_d} \left\{ s_d x_d - f(x_1, \ldots, x_d) \right\}$$

2. Work outward for $i = d - 1$ down to 1:
$$V_i(x_1, \ldots, x_{i-1}, s_i, \ldots, s_d) = \max_{x_i} \left\{ s_i x_i + V_{i+1}(x_1, \ldots, x_i, s_{i+1}, \ldots, s_d) \right\}$$

3. The final result is $f^*(s_1, \ldots, s_d) = V_1(s_1, \ldots, s_d)$

**Comparative characteristics**

Assuming equal grid resolution in primal and dual spaces ($n = m$):

1. **Dimensional decomposition**: The nested method decomposes the $d$-dimensional problem into $d$ sequential lower-dimensional maximizations. The resulting performance (generally) depends on the order in which this decomposition is performed.

2. **Computational complexity**: $\mathcal{O}(dN^{d+1})$ for the nested approach, compared to $\mathcal{O}(N^{2d})$ for the definition-based method. Each stage requires $\mathcal{O}(N^{d+1})$ operations, with $d$ total stages.

3. **Memory usage**: $\mathcal{O}(N^d)$ for the nested method.

4. **Error dependency**: Transform accuracy for both methods depends on grid resolution and therefore identical.

**Results.** Each benchmark table reports performance and accuracy metrics for various input dimensions $d$, using uniform grids with $n = m = 10$ points per dimension[11].

The column $t_{\text{Lucet}}$ (s) reports the runtime of the Nested Lucet method in seconds, while $t_{\text{Def}}$ (s) gives the runtime of the brute-force definition-based method (evaluated only for $d \leq 5$ due to computational constraints). The columns $\text{MB}_L$ and $\text{MB}_D$ indicate the peak memory usage in megabytes (MB) during execution for the Lucet and Def methods, respectively; $0.0$ reads that less than $0.05$ MB was used. The column $\max|L{-}D|$ measures the maximum absolute difference between the Lucet and Def outputs across all grid points. Lastly, `max err` captures the interpolation error of the Lucet method relative to a reference solution obtained via uniform sampling, and `RMSE` reports the root mean squared error over the same validation sample. Some columns are indicated as – since direct computations are too expensive to run.

Generally, both methods struggle with the curse of dimensionality; however, Lucet's nested LLT method is more efficient than the direct computation on the grids.

Table B1: Functions, grid domains ($10^d$ total points) and Legendre transforms.

| Function | Expression | $C$ | Legendre Transform | $D$ |
|---|---|---|---|---|
| Quadratic | $f(x) = 0.5 \sum x_i^2$ | $[-3,3]^d$ | $f^*(s) = 0.5 \sum s_i^2$ | $[-3,3]^d$ |
| Neg-Log | $f(x) = -\sum \log(x_i)$ | $[0.1, 5.0]^d$ | $f^*(s) = -d - \sum \log(-s_i)$ | $[-5.0, -0.1]^d$ |
| Quad-Linear | $f(x) = \frac{\sum x_i^2 + 1}{\sum x_i + 1}$ | $[0,3]^d$ | No explicit form | $\nabla f([0,3]^d)$ |

Table B2: Quadratic function benchmark: Lucet vs. definition-based method for different dimensions.

| $d$ | $t_{\text{Lucet}}$ (s) | $t_{\text{Def}}$ (s) | $\text{MB}_L$ | max err | RMSE |
|---|---|---|---|---|---|
| 1 | 0.001 | 0.000 | 0.0 | 5.56e−02 | 4.11e−02 |
| 2 | 0.002 | 0.002 | 0.0 | 1.11e−01 | 8.01e−02 |
| 3 | 0.012 | 0.018 | 0.0 | 1.66e−01 | 1.17e−01 |
| 4 | 0.158 | 0.614 | 0.2 | 2.11e−01 | 1.50e−01 |
| 5 | 1.981 | 83.838 | 1.5 | 2.67e−01 | 1.86e−01 |
| 6 | 25.979 | — | 15.3 | 3.27e−01 | 2.20e−01 |
| 7 | 299.697 | — | 152.6 | 3.38e−01 | 2.66e−01 |
| 8 | 3419.780 | — | 1525.9 | 3.87e−01 | 2.90e−01 |

The nested approach demonstrates substantial computational advantages as dimensionality increases, albeit at the expense of increased memory requirements. However, despite the sparse grid used, it becomes intractable for $d \geq 10$.

**Softmax-Based approximation of Legendre–Fenchel transforms**

The softmax-based approach, as proposed by Peyré, see Peyré [2020] and Blondel and Roulet [2024], offers an alternative method for approximating Legendre–Fenchel transforms through a convolution-based formulation. For a convex function $f$ defined on a bounded domain $C \subseteq \mathbb{R}^d$, the Legendre–Fenchel transform can be approximated using:

---

[11]**How we choose the dual sampling grid.** Let $\mathcal{G}_x = \{x^{(\alpha)}\}$ be the *Cartesian* primal grid. For a proper, convex, coercive function $f$ every optimal slope is a sub-gradient of $f$ at some sampled node, i.e. $\partial f(\mathcal{G}_x) = \bigcup_\alpha \partial f(x^{(\alpha)})$ contains all slopes that can ever maximise $\langle s, x \rangle - f(x)$. We therefore compute (analytically when possible, otherwise once on the discrete grid) coordinate-wise bounds $\underline{s}_i \leq \partial_{x_i} f(x^{(\alpha)}) \leq \overline{s}_i$ for all $\alpha$ and set the *dual grid* to the Cartesian product $\mathcal{G}_s = \{\underline{s}_i + k\, h_i\}_{k=0}^{M-1}$, $h_i = (\overline{s}_i - \underline{s}_i)/(M-1)$. If $\partial f(\mathcal{G}_x) \subseteq \mathcal{G}_s$ the *finite-slope-grid theorem*—a $d$-D extension of Lucet's Prop. 2 Lucet [1997]; see also Hiriart-Urruty [1980]—guarantees

$$f_{\mathcal{G}_x}^*(s) = \max_{x \in \mathcal{G}_x} \langle s, x \rangle - f(x) = f^*(s) \qquad \text{for every } s \in \mathcal{G}_s,$$

so further enlarging the slope range cannot change the transform at any sampled point—only the interpolation error between nodes depends on how fine $m$ is chosen. Example: for $f(x) = -\sum_{i=1}^d \log x_i$ on $x_i \in [0.1, 5]$ we have $\partial_{x_i} f \in [-10, -0.2]$; taking a uniform dual grid over $[-10, -0.2]$ already covers all sub-gradients and yields the exact conjugate values on that grid.

Table B3: Benchmark results for the negative log function. Lucet and definition-based methods compared across dimensions. MB: active memory (MB); $\max|L-D|$: maximum discrepancy between methods.

| $d$ | $t_{\text{Lucet}}$ (s) | $t_{\text{Def}}$ (s) | $\text{MB}_L$ | $\text{MB}_D$ | $\max|L-D|$ | max err | RMSE |
|---|---|---|---|---|---|---|---|
| 1 | 0.000 | 0.000 | 0.0 | 0.0 | 0.00e+00 | 3.95e−01 | 2.15e−01 |
| 2 | 0.001 | 0.001 | 0.0 | 0.0 | 8.88e−16 | 6.89e−01 | 3.25e−01 |
| 3 | 0.022 | 0.031 | 0.0 | 0.0 | 1.78e−15 | 9.38e−01 | 4.57e−01 |
| 4 | 0.154 | 0.659 | 0.2 | 0.5 | 3.55e−15 | 1.38e+00 | 6.36e−01 |
| 5 | 1.986 | 85.515 | 1.5 | 5.3 | 3.55e−15 | 1.48e+00 | 7.55e−01 |
| 6 | 25.952 | — | 15.3 | — | — | 1.99e+00 | 9.44e−01 |
| 7 | 305.878 | — | 152.6 | — | — | 2.19e+00 | 1.09e+00 |
| 8 | 3449.547 | — | 1525.9 | — | — | 2.26e+00 | 1.17e+00 |

Table B4: Estimated runtime comparison: definition-based vs. Lucet methods with $N = 10$ points per dimension. The definition-based method has $O(N^{2d})$ complexity while Lucet achieves $O(dN^{d+1})$.

| $d$ | Points/dim | Definition-based method (est.) | Lucet method (est.) | Speedup |
|---|---|---|---|---|
| 5 | 10 | 50.00 s | 0.62 s | 8.00e+01 |
| 6 | 10 | 1.39 hrs | 7.50 s | 6.67e+02 |
| 7 | 10 | 5.79 days | 1.46 min | 5.71e+03 |
| 8 | 10 | 1.59 yrs | 16.67 min | 5.00e+04 |
| 9 | 10 | 158.55 yrs | 3.12 hrs | 4.44e+05 |
| 10 | 10 | 15854.90 yrs | 34.72 hrs | 4.00e+06 |
| 11 | 10 | 1.59e+06 yrs | 381.94 hrs | 3.64e+07 |
| 12 | 10 | 1.59e+08 yrs | 4166.67 hrs | 3.33e+08 |
| 13 | 10 | 1.59e+10 yrs | 45138.89 hrs | 3.08e+09 |
| 14 | 10 | 1.59e+12 yrs | 486111.11 hrs | 2.86e+10 |

$$f_\varepsilon^*(s) = \varepsilon \log \left( \int_C \exp \left( \frac{\langle x, s \rangle - f(x)}{\varepsilon} \right) dx \right)$$

where $\varepsilon > 0$ is a smoothing parameter that controls the accuracy of the approximation. As $\varepsilon \to 0$, the softmax approximation $f_\varepsilon^*$ converges to the exact transform $f^*$.

This formulation can be rewritten in terms of Gaussian convolution:

$$f_\varepsilon^*(s) = Q_\varepsilon^{-1} \left( \frac{1}{Q_\varepsilon(f)} * G_\varepsilon \right)(s)$$

where: $G_\varepsilon(x) = \exp(-\frac{1}{2\varepsilon}\|x\|_2^2)$ is a Gaussian kernel, $Q_\varepsilon(f)(x) = \exp(\frac{1}{2\varepsilon}\|x\|_2^2 - \frac{1}{\varepsilon}f(x))$ and $Q_\varepsilon^{-1}(F)(s) = \frac{1}{2}\|s\|_2^2 - \varepsilon \log F(s)$.

**Implementation approaches**

Three primary techniques were employed:

- **Direct integration:** For low dimensions ($d \leq 2$), the integral is directly approximated on a grid.

- **FFT-based convolution:** For higher dimensions, the method uses the Fast Fourier Transform to compute the convolution efficiently, with complexity $\mathcal{O}(dN^d \log N)$.

- **(Quasi) Monte Carlo:** to approximate integrals. More memory efficient than integration but still expensive to iterate through dual grids or samples $\{y_i\}$.

**Empirical results**

In our experiments Monte Carlo approach showed a more stable results for performing integration for small values of $\varepsilon$, so we focus on it. Our MC implementation uses Sobol sequences[12] to generate low-discrepancy sample points, achieving better coverage of the integration domain compared to purely random sampling. Table B5 presents results for the negative logarithm function using 100,000 samples. The dual grid is the same as in LLT experiments from the previous section, see Table B3.

Experiments confirm $\mathcal{O}(\varepsilon)$ convergence to the exact Legendre transform as $\varepsilon \to 0$. Peyré's approach is suitable when an estimate of the transform is required at only one point $y$ in the dual space, but it is still too expensive to iterate over dual grids/samples, especially in high-dimensional setups: $d > 4$.

Table B5: Results of Peyré for negative log function using quasi-Monte Carlo (Sobol) with 100,000 samples across dimensions and smoothing parameter $\varepsilon$.

| $d$ | $\varepsilon$ | Sample Size | Time (s) | MB | Max Err | RMSE |
|---|---|---|---|---|---|---|
| 1 | 0.001 | qMC(100k) | 1.690 | 6.43 | 1.98e−01 | 4.43e−02 |
|   | 0.010 | qMC(100k) | 1.714 | 6.13 | 2.17e−01 | 5.33e−02 |
|   | 0.100 | qMC(100k) | 1.711 | 6.14 | 2.21e−01 | 1.30e−01 |
|   | 0.500 | qMC(100k) | 1.793 | 6.13 | 6.38e−01 | 3.55e−01 |
| 2 | 0.001 | qMC(100k) | 3.833 | 9.26 | 3.95e−01 | 6.54e−02 |
|   | 0.010 | qMC(100k) | 3.936 | 9.26 | 4.33e−01 | 8.75e−02 |
|   | 0.100 | qMC(100k) | 3.361 | 9.26 | 4.42e−01 | 2.39e−01 |
|   | 0.500 | qMC(100k) | 3.365 | 9.29 | 1.28e+00 | 5.20e−01 |
| 3 | 0.001 | qMC(100k) | 23.794 | 13.16 | 6.03e−01 | 8.73e−02 |
|   | 0.010 | qMC(100k) | 29.116 | 12.46 | 6.49e−01 | 1.21e−01 |
|   | 0.100 | qMC(100k) | 16.801 | 12.46 | 6.64e−01 | 3.47e−01 |
|   | 0.500 | qMC(100k) | 16.960 | 12.46 | 1.91e+00 | 6.56e−01 |
| 4 | 0.001 | qMC(100k) | 419.791 | 18.21 | 8.29e−01 | 1.60e−01 |
|   | 0.010 | qMC(100k) | 561.630 | 18.66 | 8.65e−01 | 1.93e−01 |
|   | 0.100 | qMC(100k) | 317.030 | 20.06 | 8.85e−01 | 4.53e−01 |
|   | 0.500 | qMC(100k) | 317.895 | 20.06 | 2.55e+00 | 7.80e−01 |

Table B6: Results of Peyré for negative entropy function using quasi-Monte Carlo (Sobol) with 100,000 samples across dimensions and smoothing parameter $\varepsilon$.

| $d$ | $\varepsilon$ | Sample Size | Time (s) | MB | Max Err | RMSE |
|---|---|---|---|---|---|---|
| 1 | 0.001 | qMC(100k) | 4.379 | 11.61 | 4.42e−01 | 1.03e−01 |
|   | 0.010 | qMC(100k) | 4.196 | 6.15 | 4.73e−01 | 1.18e−01 |
|   | 0.100 | qMC(100k) | 4.145 | 6.13 | 5.83e−01 | 2.19e−01 |
|   | 0.500 | qMC(100k) | 4.150 | 6.14 | 9.02e−01 | 4.75e−01 |
| 2 | 0.001 | qMC(100k) | 8.207 | 9.26 | 8.87e−01 | 1.57e−01 |
|   | 0.010 | qMC(100k) | 8.206 | 9.28 | 9.47e−01 | 1.87e−01 |
|   | 0.100 | qMC(100k) | 7.933 | 9.26 | 1.17e+00 | 3.75e−01 |
|   | 0.500 | qMC(100k) | 7.993 | 9.26 | 1.80e+00 | 7.04e−01 |
| 3 | 0.001 | qMC(100k) | 30.088 | 12.47 | 1.39e+00 | 2.29e−01 |
|   | 0.010 | qMC(100k) | 33.914 | 12.50 | 1.44e+00 | 2.60e−01 |
|   | 0.100 | qMC(100k) | 22.726 | 13.92 | 1.75e+00 | 5.26e−01 |
|   | 0.500 | qMC(100k) | 22.507 | 13.22 | 2.70e+00 | 8.98e−01 |
| 4 | 0.001 | qMC(100k) | 403.168 | 18.61 | 2.01e+00 | 4.10e−01 |
|   | 0.010 | qMC(100k) | 524.356 | 17.96 | 2.00e+00 | 4.33e−01 |
|   | 0.100 | qMC(100k) | 301.965 | 20.07 | 2.51e+00 | 7.17e−01 |
|   | 0.500 | qMC(100k) | 306.520 | 17.91 | 3.57e+00 | 1.07e+00 |

Our experiments showed that the method performs better for smooth functions such as negative entropy, and less effectively for functions with steep gradients like the negative logarithm.

---

[12]We tested several MC variants (basic MC / Sobol-QMC / importance / adaptive); Sobol-QMC was most stable in this setting.

**DLT: Comparing different NN architectures**

**Comparison with direct learning when the Legendre transform is known**

A critical validation of DLT is to demonstrate that it achieves state-of-the-art performance by comparing it to the direct learning of the convex conjugate in cases where the analytic form of $f^*$ is already known. For this comparison, we implemented a comprehensive benchmark that compares DLT[13] against explicit direct supervised learning across several convex functions with known convex conjugates in multiple dimensions.

**Benchmark setup**

We compare two learning approaches:

- **DLT:** trains a neural network $g_\theta$ to satisfy $g_\theta(\nabla f(x)) \approx \langle x, \nabla f(x) \rangle - f(x)$.
- **Direct:** directly trains $g_\theta$ so that $g_\theta(y) \approx f^*(y)$, which is only possible if $f^*$ is known explicitly.

The direct approach serves as our "gold standard" since it has direct access to the target function values. However, this advantage is only available in the special cases where $f^*$ has a known analytical expression—an assumption that does not hold in many practical applications. Our goal is to demonstrate that DLT matches the performance of this idealized case.

We implemented both approaches using identical neural network architectures to ensure a fair comparison:

- **MLP:** Standard multi-layer perceptron with two hidden layers (128 units each) using ReLU activations
- **MLP-ICNN:** Feed-forward network with enforced convexity using only positive weights (two hidden layers with 128 units each)
- **ResNet:** Residual network with two residual blocks (128 units each)
- **ICNN:** Input-convex neural network of Amos et al. [2017] with positive z-weights and direct x-skip connections (two hidden layers with 128 units each)

The implementation uses JAX for efficient computation with identical optimization settings for both approaches: Adam optimizer with learning rate $10^{-3}$ and batch size $128 \times d$. We employed an early stopping criterion where training terminates if the $L_2^2$ error falls below $10^{-6}$, with a maximum limit of 50,000 iterations to ensure computational feasibility across all experiments. To estimate statistical significance, we use 10 runs and report both the mean $\rho$ and standard deviation $\sigma$ of residuals achieved with direct and indirect methods.

A key consideration in our benchmark design was ensuring that sampling distributions are correctly matched between primal and dual spaces. The gradient map $\nabla f : C \to D$ transforms the distribution of points $x \in C$ into a distribution over $y = \nabla f(x) \in D$. For each function, we designed specific sampling strategies:

- **Quadratic** $f(x) = \frac{\|x\|_2^2}{2}$: Since $\nabla f(x) = x$, we sample $x \sim \mathcal{N}(0, I)$ for the implicit approach, which naturally gives $y = x \sim \mathcal{N}(0, I)$ in the dual space for the explicit approach.

- **Negative logarithm** $f(x) = -\sum_{i=1}^d \log(x_i)$: For the implicit approach, we sample $x$ uniformly in log-space from $(0.1, 10)^d$ to adequately cover the positive domain. Since $\nabla f(x) = -\frac{1}{x}$, this transforms to $y \in (-10, -0.1)^d$ in dual space. For the explicit approach, we directly sample $y$ from this same distribution.

- **Negative entropy** $f(x) = \sum_{i=1}^d x_i \log(x_i)$: Similarly, we sample $x$ uniformly in log-space for the implicit approach. With gradient $\nabla f(x) = \log(x) + 1$, this maps to $y \in (-1.3, 3.3)^d$ in dual space, which we match in our explicit approach sampling.

---

[13]DLT is an implicit approach, effectively, defined by (1.5).

This distribution matching is essential for a fair comparison, as it ensures both approaches face comparable learning challenges.

**Results and analysis**   We tested these three convex functions with known analytic conjugates across dimensions $d \in \{2, 5, 10, 20\}$:

$$\text{Quadratic:} \quad f(x) = \frac{\|x\|_2^2}{2}, \qquad\qquad\qquad f^*(y) = \frac{\|y\|_2^2}{2}$$

$$\text{Negative logarithm:} \quad f(x) = -\sum_{i=1}^{d} \log(x_i), \qquad f^*(y) = -\sum_{i=1}^{d} \log(-y_i) - d$$

$$\text{Negative entropy:} \quad f(x) = \sum_{i=1}^{d} x_i \log(x_i), \qquad f^*(y) = \sum_{i=1}^{d} \exp(y_i - 1)$$

The quadratic function $f(x) = \|x\|_2^2/2$ is particularly elegant for our comparison since it is self-conjugate ($f^* = f$) and has gradient $\nabla f(x) = x$. This means that when using DLT with samples $x \sim \mathcal{N}(0, I)$, we get $y = \nabla f(x) = x \sim \mathcal{N}(0, I)$ in the dual space as well, making the comparison especially clean.

Table B7 presents $L^2$ errors and training times for both approaches. These benchmark results reveal a key finding: DLT matches the performance of direct learning if the convex conjugate is known analytically. The $L^2$ errors for both methods are statistically *identical*.

When the dual validation samples are drawn from the training distribution, DLT learns the convex conjugate just as accurately (and almost as fast) as direct supervised learning of $f^*$ — even though it never sees the true dual values.

These results validate a key advantage of our approach: DLT achieves the same approximation accuracy as methods that require knowing the analytical form of $f^*$, without actually needing this knowledge. This makes DLT applicable to the wide range of problems where $f^*$ does not have a known closed form, while still maintaining state-of-the-art performance in cases where the Legendre transform is known.

**Experiments with non-separable functions**

To demonstrate DLT's capability beyond simple separable functions, we evaluate its performance on several challenging non-separable convex functions. Table B10 presents results across four representative function classes:

**Quadratic form with random SPD matrix**: We test $f(x) = \frac{1}{2}x^T Q x$ where $Q$ is a randomly generated symmetric positive definite matrix with controlled condition number $\kappa \approx 10^2$. The coupling through $Q$ makes this strongly non-separable, requiring the approximator to learn the full quadratic form structure.

**Exponential-minus-linear**: The function $f(x) = \exp(\langle a, x \rangle) - \langle b, x \rangle$ where $a, b \in \mathbb{S}^{d-1}$ are independently sampled unit vectors. Specifically, we sample $a_{\text{raw}}, b_{\text{raw}} \sim \mathcal{N}(0, I_d)$ and normalize them as $a = a_{\text{raw}}/\|a_{\text{raw}}\|_2$ and $b = b_{\text{raw}}/\|b_{\text{raw}}\|_2$.

**Pretrained ICNN**: We first train a 2-layer ICNN (128 hidden units per layer) for 50,000 iterations to approximate a target quadratic function, then freeze it and use it as the convex function $f$. This represents a realistic scenario where the target function itself is a learned model from data.

**Coupled Soft-plus**: The function $f(x) = \sum_{i<j} \log(1 + \exp(x_i + x_j))$ explicitly couples all pairs of variables with $O(d^2)$ interaction terms. This severely challenges classical methods due to the combinatorial explosion of terms.

**Experimental setup.**   All experiments employ ResNet approximators with 2 blocks of 128 hidden units each, trained using the DLT framework. We train for 120,000 iterations with a learning rate of $10^{-3}$ and Huber loss ($\delta = 1.0$) to ensure robust convergence. For each function class, we apply domain-specific whitening transformations. Training samples $x$ are drawn from appropriate domains

Table B7: Comparison of DLT with direct learning for known Legendre transforms

| Function | $d$ | Model | $L^2$ Error | | Time (s) | |
|---|---|---|---|---|---|---|
| | | | DLT | Dir | DLT | Dir |
| Quadratic | 10 | MLP | 2.96e-05 | 2.86e-05 | 85.5 | 84.4 |
| | | MLP-ICNN | 6.21e+00 | 3.15e+01 | 87.4 | 76.9 |
| | | ResNet | 2.43e-05 | 3.00e-05 | 106.8 | 101.2 |
| | | ICNN | 1.79e-02 | 1.23e-02 | 92.7 | 93.0 |
| | 20 | MLP | 2.28e-04 | 3.41e-04 | 84.9 | 84.0 |
| | | MLP-ICNN | 1.20e+02 | 1.13e+02 | 71.3 | 82.6 |
| | | ResNet | 2.53e-04 | 2.32e-04 | 107.7 | 104.9 |
| | | ICNN | 3.45e-02 | 3.48e-02 | 95.1 | 93.3 |
| | 50 | MLP | 3.55e-03 | 3.59e-03 | 87.5 | 86.6 |
| | | MLP-ICNN | 6.46e+02 | 6.46e+02 | 87.3 | 76.6 |
| | | ResNet | 2.92e-03 | 3.25e-03 | 123.8 | 119.7 |
| | | ICNN | 7.18e-02 | 6.66e-02 | 97.4 | 97.8 |
| Neg. Log | 10 | MLP | 6.75e-04 | 8.43e-04 | 94.1 | 94.4 |
| | | MLP-ICNN | 4.23e+00 | 4.23e+00 | 63.7 | 62.4 |
| | | ResNet | 7.34e-04 | 9.22e-04 | 108.7 | 111.0 |
| | | ICNN | 6.47e+00 | 4.22e+00 | 54.1 | 65.4 |
| | 20 | MLP | 4.31e-03 | 3.60e-03 | 93.8 | 95.8 |
| | | MLP-ICNN | 7.43e+00 | 7.16e+00 | 86.2 | 96.8 |
| | | ResNet | 5.36e-03 | 6.20e-03 | 114.0 | 115.2 |
| | | ICNN | 8.75e+00 | 8.69e+00 | 71.4 | 63.5 |
| | 50 | MLP | 3.81e-01 | 2.81e-01 | 97.5 | 97.8 |
| | | MLP-ICNN | 2.03e+01 | 2.03e+01 | 67.3 | 70.4 |
| | | ResNet | 1.05e-01 | 1.23e-01 | 133.5 | 130.7 |
| | | ICNN | 2.03e+01 | 2.06e+01 | 73.5 | 61.9 |
| Neg. Entropy | 10 | MLP | 1.06e-03 | 1.26e-03 | 95.6 | 100.3 |
| | | MLP-ICNN | 4.56e+00 | 4.01e+00 | 92.8 | 101.0 |
| | | ResNet | 6.62e-04 | 7.41e-04 | 109.8 | 115.7 |
| | | ICNN | 1.31e-02 | 1.74e-02 | 98.8 | 107.2 |
| | 20 | MLP | 5.88e-03 | 7.83e-03 | 94.5 | 99.8 |
| | | MLP-ICNN | 2.80e+01 | 2.82e+01 | 92.7 | 102.2 |
| | | ResNet | 3.88e-03 | 4.24e-03 | 113.7 | 118.6 |
| | | ICNN | 3.68e+01 | 3.67e+01 | 68.1 | 72.3 |
| | 50 | MLP | 2.71e-01 | 3.91e-01 | 99.6 | 100.8 |
| | | MLP-ICNN | 7.83e+01 | 7.87e+01 | 97.5 | 104.6 |
| | | ResNet | 6.93e-02 | 7.27e-02 | 132.8 | 135.0 |
| | | ICNN | 9.39e+01 | 9.39e+01 | 59.8 | 69.1 |

with truncated normal or uniform distribution (see Table B9), with batch sizes adaptively set to 25% of the dimension ($B = \lfloor 0.25d \rfloor$). We evaluate performance at dimensions $d \in \{10, 20, 50\}$ and report results averaged over 5 independent trials with different random seeds.

**Results.** The results demonstrate that DLT successfully learns all four non-separable function classes, achieving low RMSE errors across different dimensions. The quadratic SPD function shows the best performance with errors decreasing as dimension increases (from $1.15 \times 10^{-2}$ at $d = 10$ to $2.01 \times 10^{-3}$ at $d = 50$), demonstrating DLT's effectiveness in learning structured covariance patterns. The exponential-minus-linear function presents greater challenges due to its unbounded gradient domain, yet DLT maintains reasonable accuracy. The pretrained ICNN case validates DLT's ability to approximate learned neural network functions.

Table B8: Comparison of DLT with direct learning for known Legendre transforms

| Function | $d$ | Model | $L^2$ Error | | Time (s) | |
|---|---|---|---|---|---|---|
| | | | DLT | Dir | DLT | Dir |
| Quadratic | 50 | MLP | 3.55e-03 | 3.59e-03 | 87.5 | 86.6 |
| | | MLP-ICNN | 6.46e+02 | 6.46e+02 | 87.3 | 76.6 |
| | | ResNet | 2.92e-03 | 3.25e-03 | 123.8 | 119.7 |
| | | ICNN | 7.18e-02 | 6.66e-02 | 97.4 | 97.8 |
| | 100 | MLP | 2.77e-02 | 2.46e-02 | 95.0 | 95.3 |
| | | MLP-ICNN | 2.54e+03 | 2.59e+03 | 96.8 | 87.4 |
| | | ResNet | 1.88e-02 | 1.91e-02 | 136.3 | 132.0 |
| | | ICNN | 9.59e-02 | 9.06e-02 | 105.7 | 105.5 |
| | 200 | MLP | 1.43e-01 | 1.50e-01 | 113.5 | 112.7 |
| | | MLP-ICNN | 1.01e+04 | 1.01e+04 | 97.6 | 113.7 |
| | | ResNet | 9.53e-02 | 9.40e-02 | 163.1 | 164.4 |
| | | ICNN | 4.01e-01 | 3.73e-01 | 124.1 | 125.8 |
| Neg. Log | 50 | MLP | 3.81e-01 | 2.81e-01 | 97.5 | 97.8 |
| | | MLP-ICNN | 2.03e+01 | 2.03e+01 | 67.3 | 70.4 |
| | | ResNet | 1.05e-01 | 1.23e-01 | 133.5 | 130.7 |
| | | ICNN | 2.03e+01 | 2.06e+01 | 73.5 | 61.9 |
| | 100 | MLP | 8.56e+00 | 6.67e+00 | 106.2 | 106.1 |
| | | MLP-ICNN | 4.00e+01 | 3.99e+01 | 73.0 | 66.2 |
| | | ResNet | 1.38e+00 | 1.93e+00 | 141.5 | 142.3 |
| | | ICNN | 3.99e+01 | 3.98e+01 | 80.3 | 71.1 |
| | 200 | MLP | 3.29e+01 | 2.68e+01 | 124.7 | 125.1 |
| | | MLP-ICNN | 8.21e+01 | 8.20e+01 | 68.9 | 78.7 |
| | | ResNet | 2.92e+01 | 2.99e+01 | 168.7 | 174.1 |
| | | ICNN | 1.70e+01 | 1.68e+01 | 131.2 | 137.0 |
| Neg. Entropy | 50 | MLP | 2.71e-01 | 3.91e-01 | 99.6 | 100.8 |
| | | MLP-ICNN | 7.83e+01 | 7.87e+01 | 97.5 | 104.6 |
| | | ResNet | 6.93e-02 | 7.27e-02 | 132.8 | 135.0 |
| | | ICNN | 9.39e+01 | 9.39e+01 | 59.8 | 69.1 |
| | 100 | MLP | 1.77e+01 | 1.65e+01 | 106.5 | 108.7 |
| | | MLP-ICNN | 1.45e+02 | 1.44e+02 | 103.5 | 111.2 |
| | | ResNet | 1.53e+00 | 1.32e+00 | 142.0 | 144.2 |
| | | ICNN | 1.93e+02 | 1.93e+02 | 62.9 | 79.9 |
| | 200 | MLP | 4.92e+01 | 5.07e+01 | 124.2 | 127.3 |
| | | MLP-ICNN | 2.87e+02 | 2.87e+02 | 125.3 | 133.3 |
| | | ResNet | 5.08e+01 | 4.02e+01 | 94.2 | 175.4 |
| | | ICNN | 9.43e+01 | 9.30e+01 | 131.9 | 139.2 |

Notably, the coupled soft-plus function requires approximately $2\times$ longer training time at $d = 20$ and $d = 50$ due to its $O(d^2)$ pairwise interaction terms and longer gradient evaluation step. The relative RMSE remains below 2% for most cases, confirming that DLT provides accurate approximations suitable for downstream optimization tasks. These experiments establish that DLT with simple ResNet architectures can effectively handle diverse non-separable convex functions that would be intractable for classical decomposition methods.

**Alternative NN approaches to computing Legendre transforms**

DLT approximates the convex conjugate $f^*$ on $D = \nabla f(C)$ by leveraging the exact Fenchel–Young identity at gradient-mapped points. The key insight is that for any $x \in C$ with $y = \nabla f(x)$, the identity $f^*(y) = \langle x, y \rangle - f(x)$ holds exactly.

Table B9: Test functions with sampling domains. $C$ indicates the effective training region (primal domain), $D = \nabla f(C)$ is the corresponding dual domain for gradient samples.

| Function | Expression | $C$ (Primal) | $D$ (Dual) |
|---|---|---|---|
| Quadratic SPD | $f(x) = \frac{1}{2}x^\top Q x$ (random SPD, $\|Q\|_2 \approx 1$) | $[-3,3]^d$ (std. normal $\pm 3\sigma$) | $[-3\|Q\|_2, 3\|Q\|_2]^d$ ($\approx [-3,3]^d$ when normalized) |
| Exp-minus-Lin | $f(x) = e^{\langle a,x\rangle} - \langle b,x\rangle$ ($\|a\|_2 = \|b\|_2 = 1$) | $[-3,3]^d$ (std. normal $\pm 3\sigma$) | $[e^{-3\sqrt{d}}-1, e^{3\sqrt{d}}-1]^d$ |
| ICNN (2-layer) | $f(x) = \text{ICNN}_{128,128}(x)$ (pretrained to quadratic) | $[-3,3]^d$ | Learned, data-dependent ($\approx [-3,3]^d$) |
| Soft-plus pairs | $f(x) = \sum_{i<j} \log(1+e^{x_i+x_j})$ (pairwise coupling) | $[-1.5,1.5]^d$ (uniform ) | $\approx [0, d-1]^d$ |

Table B10: Performance of DLT with ResNet approximators for non-separable functions. Absolute RMSE, relative RMSE (normalized by function scale) and training times are reported as mean $\pm$ standard deviation over 5 independent trials.

| Function | $d$ | RMSE | Relative RMSE | Time (s) |
|---|---|---|---|---|
| Quadratic SPD | 10 | 1.15e-2 $\pm$ 3.7e-3 | 1.05e-2 $\pm$ 3.9e-3 | 584 $\pm$ 9 |
| | 20 | 5.13e-3 $\pm$ 1.1e-3 | 3.98e-3 $\pm$ 6.3e-4 | 1016 $\pm$ 10 |
| | 50 | 2.01e-3 $\pm$ 2.0e-3 | 1.05e-3 $\pm$ 1.1e-3 | 2709 $\pm$ 2 |
| Exp-minus-Lin | 10 | 9.18e-2 $\pm$ 3.6e-2 | 2.50e-2 $\pm$ 1.0e-2 | 599 $\pm$ 9 |
| | 20 | 5.78e-2 $\pm$ 3.3e-2 | 1.60e-2 $\pm$ 8.8e-3 | 1035 $\pm$ 14 |
| | 50 | 5.41e-2 $\pm$ 4.7e-2 | 1.41e-2 $\pm$ 1.2e-2 | 2712 $\pm$ 0.2 |
| ICNN (2-layer) | 10 | 2.37e-2 $\pm$ 7.1e-3 | 6.17e-3 $\pm$ 1.4e-3 | 706 $\pm$ 7 |
| | 20 | 1.92e-2 $\pm$ 5.3e-3 | 6.06e-3 $\pm$ 1.6e-3 | 1200 $\pm$ 10 |
| | 50 | 1.13e-2 $\pm$ 2.4e-3 | 3.69e-3 $\pm$ 8.1e-4 | 2759 $\pm$ 1 |
| Soft-plus pairs | 10 | 1.14e-2 $\pm$ 7.9e-3 | 5.21e-3 $\pm$ 3.6e-3 | 965 $\pm$ 9 |
| | 20 | 2.63e-2 $\pm$ 9.4e-3 | 4.14e-3 $\pm$ 1.5e-3 | 2181 $\pm$ 11 |
| | 50 | 1.25e-1 $\pm$ 4.3e-2 | 4.95e-3 $\pm$ 1.7e-3 | 5498 $\pm$ 13 |

We train a neural network $g_\theta : D \to \mathbb{R}$ by minimizing the loss:

$$[g_\theta(\nabla f(x)) + f(x) - \langle x, \nabla f(x)\rangle]^2 \tag{DLT}$$

where $x \in C$ is sampled to ensure $y = \nabla f(x)$ adequately covers $D$. Crucially, this loss equals $(g_\theta(y) - f^*(y))^2$ for points on the manifold $y = \nabla f(x)$, providing an immediate accuracy certificate through the training residual itself.

In contrast, one can consider a proxy method by using an approximate target:

$$f^*(y) \approx \langle h_\vartheta(y), y\rangle - f(h_\vartheta(y)), \quad \text{where } h_\vartheta \approx (\nabla f)^{-1} \tag{Proxy}$$

This introduces potential bias since $h_\vartheta$ is only an approximation of the true inverse gradient. More precisely, if a model is trained to the proxy target $t_h(y) = \langle h_\vartheta(y), y\rangle - f((y))$, then

$$g_\theta(y) - f^*(y) = \underbrace{g_\theta(y) - t_h(y)}_{\text{optim. error}} + \underbrace{t_h(y) - f^*(y)}_{\text{bias from } h_\vartheta}.$$

Even if optimization drives the first term down, the *certificate* still reflects the second term, which is zero only when $h_\vartheta = (\nabla f)^{-1}$ on $D$. While proxy methods can achieve low training loss on their approximate targets, we evaluate all methods using our DLT certificate (DLT), which measures the true approximation error regardless of the training approach used.

DLT provides several critical advantages over the Proxy method:

1. **Guaranteed convexity**: When using an ICNN architecture, DLT guarantees the convexity of the output $g_\theta$ since the training target is the true convex conjugate $f^*$. The Proxy method

loses this guarantee because even with an ICNN, the approximator learns a biased target $\langle h_\vartheta(y), y \rangle - f(h_\vartheta(y))$ where $h_\vartheta$ may not be the true inverse gradient.

2. **No accuracy bottleneck**: DLT's accuracy is limited only by the neural network's approximation capacity, not by an intermediate operator approximation. The Proxy method's final accuracy is fundamentally bottlenecked by the precision of $h_\vartheta \approx (\nabla f)^{-1}$, regardless of the main network's capacity.

3. **Efficient sampling**: DLT uses the exact gradient mapping $\nabla f$ for sampling, which is computationally efficient and exact. The Proxy method requires every new sample to pass through the learned inverse network $h_\vartheta$, adding computational overhead and potential approximation errors at inference time.

4. **Training overhead**: While both methods may use an inverse mapping for sampling control, the Proxy method critically depends on the quality of this mapping for its training targets, requiring more careful and time-consuming pretraining.

Table B11: DLT with approximate-inverse sampling vs. proxy. $h_\vartheta$ RMSE measures prelearned inverse quality; $t_h$: inverse pretraining time.

| Function | $d$ | Method | Train (s) | RMSE | $h_\vartheta$ RMSE | $t_h$ (s) |
|---|---|---|---|---|---|---|
| Quadratic | 5 | DLT + inv.sampling | 113.95 | 8.29e−03 | 8.23e−06 | 287.8 |
| | | Proxy | 104.60 | 1.49e−02 | — | 287.8 |
| | 10 | DLT + inv.sampling | 222.05 | 2.02e−02 | 7.41e−05 | 468.4 |
| | | Proxy | 209.22 | 2.10e−02 | — | 468.4 |
| Neg. Log | 5 | DLT + inv.sampling | 107.96 | 3.98e−02 | 5.79e−06 | 229.7 |
| | | Proxy | 104.28 | 4.21e−02 | — | 229.7 |
| | 10 | DLT + inv.sampling | 209.41 | 1.28e−01 | 8.03e−05 | 390.2 |
| | | Proxy | 204.96 | 1.45e−01 | — | 390.2 |
| Neg. Entropy | 5 | DLT + inv.sampling | 110.70 | 3.27e−02 | 3.27e−05 | 326.5 |
| | | Proxy | 106.79 | 3.25e−02 | — | 326.5 |
| | 10 | DLT + inv.sampling | 221.21 | 3.53e−02 | 1.88e−04 | 475.0 |
| | | Proxy | 212.85 | 3.64e−02 | — | 475.0 |

Table B11 reports numerical results[14] for the functions from Table B4, comparing DLT with approximate-inverse sampling with the inverse gradient proxy baseline for dimensions $d \in \{5, 10\}$. Both methods share the same inverse pretraining time $t_h$, which ranges from approximately 4 to 8 minutes depending on the function and dimension. We learn the inverse mapping $h_\vartheta$ to high precision ($h_\vartheta$ RMSE between $10^{-6}$ and $10^{-4}$). As a result, the certificate RMSE—which measures the true approximation error of $f^*$—becomes comparable in terms of training times between the two methods.

**Handling out-of-domain inverse mappings.** When using an approximate inverse mapping $h_\vartheta : D \to \mathbb{R}^d$ for sampling, some points $h_\vartheta(y)$ may fall outside the constraint set $C$. Two approaches exist for handling these cases: (i) adding a penalty term to the loss function that penalizes violations of $h_\vartheta(y) \notin C$, or (ii) discarding such samples from the training batch. While the penalty approach could

---

[14]Both methods employ ResNet architectures with 2 blocks of 128 hidden units and GELU activations. The key is a two-stage procedure: first, we pretrain an approximate inverse mapping $h_{\text{samp}}$ using an autoencoder-style cycle that maps $y \to z = g(y) \to \hat{x} = h_\vartheta(z) \to \nabla f(\hat{x})$, ensuring $\hat{x}$ remains within the primal domain $C$ through projection. This pretraining phase uses 200,000 steps for $d = 5$ and 400,000 steps for $d = 20$ with larger batch sizes ($1.01\times$ the streaming batch). In the main training phase, both methods use streaming mini-batches with fresh samples $y \sim \text{Uniform}(D)$ drawn at each step, where the dual domain $D$ is carefully matched to the primal domain $C$ via the gradient mapping. To ensure numerical stability, we apply interior clipping with relative tolerance $\varepsilon = 10^{-6}$ to keep gradients strictly within valid domains. The streaming batch size $N$ is set explicitly (600 for $d = 5$, 1280 for $d = 20$) and scaled by $0.99\times$ for both methods, with training steps compensated proportionally to maintain constant total sample complexity (30,000 base steps for $d = 5$, 100,000 for $d = 20$). Training employs AdamW optimization with learning rate $10^{-3}$, weight decay $10^{-6}$, and exponential decay schedule (halving every 20,000 steps). Out-of-bounds gradient samples are dropped. Performance is evaluated on a test set of 5,000 $(x, y)$ pairs measuring both mean squared error and relative $L_2$ error $\|\hat{g} - f^*\| / \|f^*\|$ of the conjugate function approximation.

theoretically guide the network toward valid mappings, it introduces additional hyperparameters (penalty weight, barrier function choice) and may distort the training dynamics by mixing two objectives—approximating the inverse and satisfying constraints. We adopt the discarding approach as it maintains the purity of the training objective.

**Hamilton–Jacobi equation and time-parameterized inverse gradient sampling**

The Hamilton–Jacobi (HJ) equation is a cornerstone of classical mechanics, optimal control theory, and mathematical physics. It is given by the first-order nonlinear partial differential equation

$$\partial_t u + H(\nabla_x u(x,t)) = 0,$$

where $u(x,t)$ represents the action (value) function, and $H$ denotes the Hamiltonian associated with a physical system or control problem.

Consider the Cauchy problem:

$$\begin{cases} \partial_t u + H(\nabla_x u(x,t)) = 0, & (x,t) \in Q, \\ u(x,0) = g(x), & x \in \mathbb{R}^d, \end{cases}$$

where $Q = \mathbb{R}^d \times (0, \infty)$.

**The Hopf formula and function definitions**

The Hopf formula provides an explicit solution to the Hamilton–Jacobi equation:

$$u(x,t) = (g^* + tH)^*(x) = \sup_{p \in \mathbb{R}^d} \{\langle x, p \rangle - g^*(p) - tH(p)\} \tag{B1}$$

For our computational experiments, we consider two classes of functions that exhibit different mathematical properties and computational challenges.

**Quadratic functions.** These represent the classical case with explicit analytical solutions:

- Initial condition: $g(x) = \frac{1}{2} \sum_{i=1}^d x_i^2$
- Conjugate: $g^*(p) = \frac{1}{2} \sum_{i=1}^d p_i^2$
- Hamiltonian: $H(p) = \frac{1}{2} \sum_{i=1}^d p_i^2$

In this case, the analytical solution is $u(x,t) = \frac{\sum_{i=1}^d x_i^2}{2(1+t)}$.

**Exponential functions.** These provide a more challenging test case:

- Initial condition: $g(x) = \sum_{i=1}^d e^{x_i}$
- Hamiltonian: $H(p) = \sum_{i=1}^d e^{p_i}$
- Conjugate: $g^*(p) = \sum_{i=1}^d (p_i \log p_i - p_i)$ (when $p_i > 0$)

To approximate the HJ solution $u(x,t)$ we use a time-parameterized network $u_\theta(x,t)$ with input $(x,t)$.

**Time-parameterized inverse gradient sampling**

Denote $v(p,t) := g^*(p) + t\,H(p)$. Then the Hopf representation can be written as

$$u(x,t) = v(\cdot,t)^*(x) = \sup_{p \in \mathbb{R}^d} \{\langle x, p \rangle - v(p,t)\}.$$

Whenever $v(\cdot,t)$ is differentiable (and, e.g., strictly convex so the maximizer is unique), the maximizer $p = p(x,t)$ is characterized by the first-order optimality condition

$$x = \nabla_p v(p,t) = \nabla g^*(p) + t\,\nabla H(p), \qquad \text{hence} \qquad p = \nabla_x u(x,t) = (\nabla_p v(\cdot,t))^{-1}(x).$$

Time-DLT therefore requires samples of pairs $(x, p)$ satisfying this relation. In the case where $H$ and $g$ are quadratic one has $\nabla_p v(p, t) = (1 + t)p$, so $x = (1 + t)p$ (equivalently $p = x/(1 + t)$), which allows direct sampling. In non-quadratic cases the inverse map $(\nabla_p v(\cdot, t))^{-1}$ is not available in closed form; to generate samples with a prescribed distribution in $x$-space we learn a time-parameterized inverse-gradient network $\Psi_\phi(x, t) \approx (\nabla_p v(\cdot, t))^{-1}(x)$. Similar to Section 5 we train $\Psi_\phi$ using $\|\nabla_p v(\Psi_\phi(x, t), t) - x\|_2^2$ and $\|\Psi_\phi(\nabla_p v(p, t), t) - p\|_2^2$ as loss funcitons, which only require evaluating $\nabla_p v = \nabla g^* + t\nabla H$. In the implementation, this provides the inverse-gradient sampling step $p \approx \Psi_\phi(x, t)$ (and in the quadratic case the explicit formula $p = x/(1 + t)$) used to form training triplets $(x, p, t)$.

**Time-DLT implementation**

Time-DLT directly approximates the Hopf formula through with neural network $u_\theta(x, t)$ that learns:

$$u_\theta(x, t) \approx u(x, t) = \sup_p \left\{ \langle x, p \rangle - g^*(p) - tH(p) \right\}.$$

The training process minimizes the Legendre transform residual:

$$\mathcal{L} = \mathbb{E}_{(x,p,t)} \left[ \left| u_\theta(x, t) - (\langle x, p \rangle - g^*(p) - tH(p)) \right|^2 \right].$$

The sampling of triplets $(x, p, t)$ depends on the function type: analytical formulas for quadratic functions and the trained inverse network for exponential functions. This approach preserves the structure of the Hopf formula while ensuring computational tractability.

**Experimental framework**

We evaluate both methods across increasing dimensions and function complexities. We sample $(x, t) \sim \text{Unif}\left([x_{\min}, x_{\max}]^d \times [0, t_{max}]\right)$, with $x_{\min} = -2, x_{max} = 2, t_{max} = 3$ and report in Tables B12 and B13 the resulted RMSE and PDE residuals across different time moments $t \in \{0.1, 0.5, 1.0, 2.0\}$.

**Problem configurations.** We evaluate both methods across the following configurations:

- **Dimensions:** $d \in \{2, 5, 10\}$ to assess scalability

- **Function combinations:**
  - Quadratic–Quadratic: baseline with known analytical solution
  - Quadratic–Exponential: Hamiltonian $H$ is exponential
  - Exponential–Quadratic: initial condition $g$ is exponential

**Neural network architecture** Both methods employ time-parameterized multilayer perceptrons (MLPs):

- **Input:** $(x, t) \in \mathbb{R}^d \times \mathbb{R}^+$

- **Architecture:** $(d + 1) \to 64 \to 64 \to 1$

- **Activation:** $\tanh$

- **Optimizer:** Adam with learning rate $\eta = 10^{-3}$

For Time-DLT, the inverse gradient network $h_\phi$ uses:

- **Input:** $(p, t) \in \mathbb{R}^d \times \mathbb{R}^+$

- **Architecture:** $(d + 1) \to 64 \to 64 \to d$

- **Two-phase training:** 40% of epochs for $h_\phi$, 60% for $F_\theta$

**Implementation details**

**Ground truth computation.** To assess accuracy when an explicit solution is unavailable, we compute reference solutions by evaluating for each point $(x, t)$ a reference value by solving the inner optimization over $p$ in the Hopf representation (B1):

- Low dimensions ($d \leq 2$): Scipy's L-BFGS-B algorithm
- Higher dimensions ($2 < d \leq 5$): PyTorch gradient-based optimization with Adam

**Training specifications.**

- **Batch size:** 100 samples per iteration
- **Training epochs:** 1500 for $d = 2$, scaling to 3000 for $d = 10$
- **Sampling strategy:** Adaptive mixing of initial condition and interior points
- **Regularization:** $\lambda = 100$ for initial condition enforcement
- **Gradient clipping:** Maximum norm of 10.0 for stability

**Evaluation metrics**

We assess performance using three complementary metrics:

1. $L^2$ **error against ground truth:**

$$\varepsilon_{L^2} = \sqrt{\frac{1}{N} \sum_{i=1}^{N} |u_\theta(x_i, t) - u_{\text{ref}}(x_i, t)|^2}$$

2. **PDE residual:**

$$\varepsilon_{\text{PDE}} = \sqrt{\mathbb{E}_{(x,t)} \left[|\partial_t u_\theta + H(\nabla_x u_\theta)|^2\right]}$$

3. **Initial condition error:**

$$\varepsilon_{\text{IC}} = \sqrt{\mathbb{E}_x \left[|u_\theta(x, 0) - g(x)|^2\right]}$$

The results demonstrate several key findings:

1. **Accuracy:** Time-DLT consistently achieves significantly lower errors compared to DGM across all problem configurations.
2. **Scaling with dimension:** Both methods show increased errors as dimension increases, but Time-DLT maintains superior performance even in higher dimensions.

The Time-DLT approach demonstrates that by explicitly modeling the time-parameterized convex conjugate $(g^* + tH)^*$, we can achieve more accurate and efficient solutions compared to direct PDE approximation methods. This approach is particularly effective for high-dimensional problems where $g^*$ is known in closed form. If DLT is used to compute $g^*$ as well, errors from such a double Legendre transform might accumulate, potentially degrading overall approximation accuracy.

Table B12: Comparison of DGM and Time-DLT on the Hamilton–Jacobi equation with quadratic initial condition and Hamiltonian

| $d$ | $t$ | DGM | | | Time-DLT | | | Err. Ratio | |
|---|---|---|---|---|---|---|---|---|---|
| | | $L^2$ Error | PDE Res. | Time (s) | $L^2$ Error | PDE Res. | Time (s) | Mean | $\sigma$ |
| 2 | 0.1 | 8.88e-3 | 9.63e-2 | 80.66 | 3.43e-3 | 6.81e-1 | 56.29 | 2.69 | 0.46 |
| | 0.5 | 2.28e-2 | 4.98e-2 | | 3.89e-3 | 3.67e-1 | | 7.53 | 1.64 |
| | 1.0 | 3.92e-2 | 3.36e-2 | | 3.23e-3 | 2.07e-1 | | 11.38 | 2.43 |
| | 2.0 | 4.20e-2 | 1.93e-2 | | 2.65e-3 | 9.22e-2 | | 17.39 | 6.11 |
| 5 | 0.1 | 3.34e-2 | 2.47e-1 | 82.75 | 2.81e-2 | 1.10e+0 | 57.42 | 1.24 | 0.13 |
| | 0.5 | 1.09e-1 | 1.27e-1 | | 1.98e-2 | 5.87e-1 | | 5.55 | 0.85 |
| | 1.0 | 1.55e-1 | 8.27e-2 | | 1.39e-2 | 3.31e-1 | | 10.29 | 1.77 |
| | 2.0 | 1.16e-1 | 3.97e-2 | | 1.08e-2 | 1.49e-1 | | 13.92 | 4.25 |
| 10 | 0.1 | 8.90e-2 | 4.82e-1 | 82.40 | 8.46e-2 | 1.60e+0 | 58.76 | 0.95 | 0.09 |
| | 0.5 | 2.14e-1 | 3.04e-1 | | 5.25e-2 | 9.17e-1 | | 3.69 | 0.44 |
| | 1.0 | 2.26e-1 | 1.85e-1 | | 3.68e-2 | 5.12e-1 | | 7.32 | 1.30 |
| | 2.0 | 1.64e-1 | 8.05e-2 | | 2.72e-2 | 2.27e-1 | | 8.64 | 2.57 |
| 20 | 0.1 | 2.53e-1 | 9.51e-1 | 87.39 | 2.43e-1 | 2.25e+0 | 88.85 | 1.03 | 0.09 |
| | 0.5 | 3.78e-1 | 6.83e-1 | | 1.49e-1 | 1.36e+0 | | 2.96 | 0.34 |
| | 1.0 | 3.87e-1 | 4.03e-1 | | 1.13e-1 | 7.50e-1 | | 4.22 | 0.97 |
| | 2.0 | 4.39e-1 | 1.81e-1 | | 7.36e-2 | 3.24e-1 | | 6.14 | 0.96 |
| 30 | 0.1 | 3.45e-1 | 1.18e+0 | 87.18 | 4.29e-1 | 2.69e+0 | 192.62 | 0.83 | 0.12 |
| | 0.5 | 6.40e-1 | 8.85e-1 | | 2.60e-1 | 1.82e+0 | | 2.63 | 0.34 |
| | 1.0 | 1.01e+0 | 4.68e-1 | | 1.90e-1 | 9.69e-1 | | 5.15 | 1.23 |
| | 2.0 | 1.21e+0 | 2.56e-1 | | 1.28e-1 | 3.98e-1 | | 9.00 | 2.21 |

Notes: $L^2$ Error represents the mean squared difference between predicted solutions and the analytical solution $u(x,t) = \frac{\sum x_i^2}{2(1+t)}$. HJ Error measures the residual of the Hamilton–Jacobi equation $|\partial_t u + H(\nabla_x u)|$.

Table B13: Comparison of DGM and Time-DLT on the Hamilton–Jacobi equation with quadratic initial condition and exponential Hamiltonian

| $d$ | $t$ | DGM | | | Time-DLT | | | Error Ratio |
|---|---|---|---|---|---|---|---|---|
| | | $L^2$ Error | PDE Residual | Time (s) | $L^2$ Error | PDE Residual | Time (s) | DGM / DLT |
| 2 | 0.1 | 1.46e-01 | 2.25e-01 | 142.40 | 6.88e-02 | 2.24e-01 | 95.35 | 2.12 |
| | 0.5 | 1.30e+00 | 1.27e-01 | | 1.29e+00 | 3.29e-02 | | 1.01 |
| | 1.0 | 3.25e+00 | 8.21e-02 | | 3.21e+00 | 7.19e-02 | | 1.01 |
| | 2.0 | 7.70e+00 | 6.48e-02 | | 7.69e+00 | 6.87e-02 | | 1.00 |
| 5 | 0.1 | 3.18e-01 | 5.37e-01 | 215.76 | 1.14e+00 | 1.43e+00 | 145.49 | 0.28 |
| | 0.5 | 5.53e-01 | 4.16e-01 | | 4.94e-01 | 9.03e-01 | | 1.12 |
| | 1.0 | 1.32e+00 | 3.67e-01 | | 4.69e-01 | 4.55e-01 | | 2.80 |
| | 2.0 | 2.64e+00 | 6.53e-01 | | 4.90e-01 | 2.70e-01 | | 5.39 |

**Notes:** $L^2$ Error quantifies the mean squared deviation from the ground truth solution. PDE Residual is the residual of the Hamilton–Jacobi equation $|\partial_t u + H(\nabla_x u)|$.

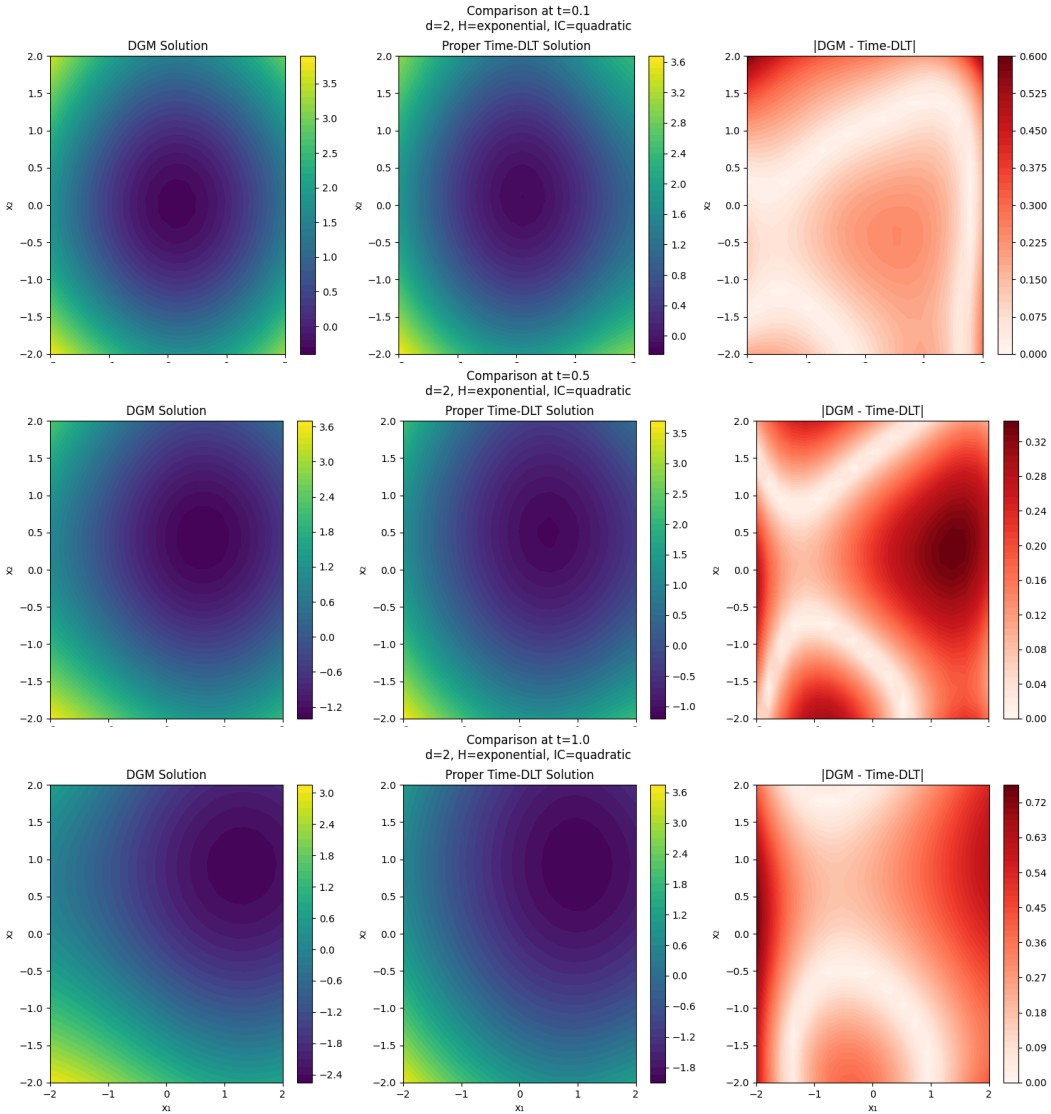

Figure B1: Comparison of DGM and Time-DLT at three different times $t = 0.1, 0.5, 1$ for $d = 2$.

**Approximate inverse sampling: additional experiments**

In this section, we consider two more examples of approximate inverse sampling in the context of the Deep Legendre Transform.

The first example, when approximate inverse sampling might be particularly useful, is when the target distribution $\nu$ is concentrated around certain points $\{y_i\}_{i=1}^n \in D$ of particular interest. Then, on the one hand, one has to sample $x$ from a distribution $\mu$, such that $\nabla f \circ \mu$ has high density near all the points $\{y_i\}_{i=1}^n \in D$. On the other hand, if the points are distant enough, it might be inefficient to try to learn $f^*$ on the whole $D = \nabla f(C)$ given that we are interested in higher precision in the neighborhood of $\{y_i\}_{i=1}^n \in D$. We illustrate that approximate inverse sampling can solve both problems, efficiently sampling (approximately) to the target distribution $\nu$.

In the second example, we explore the ability of KANs to find symbolic representations in low-dimensional setups with and without approximate inverse sampling. The setup is similar to that of Section 6.1 of the main part. Surprisingly, KANs managed to find correct symbolic expression even with a quite skewed sample.

**The case of a target distribution concentrated around specific points**

We consider the case of the multidimensional negative entropy

$$f(x) = \sum_{i=1}^{d} x_i \log(x_i)$$

defined on the positive orthant $C = \{x \in \mathbb{R}^d : x_i > 0 \text{ for all } i\}$. It is straightforward to check that $f^*(x) = \sum_{i=1}^{d} \exp(x_i - 1)$.

Consider two points in $\mathbb{R}^d$ : a point $y$ with coordinates $y_i = 1$, for all $i$, and a point $z$ with all coordinates equal to $4$. The target distribution $\nu$ is modeled as a fair mixture of normal distributions $\mathcal{N}(y, 10^{-3} I_d)$ and $\mathcal{N}(z, 10^{-3} I_d)$.

Figures B3 and B3 illustrate the process of learning the inverse gradient mapping in this setup.

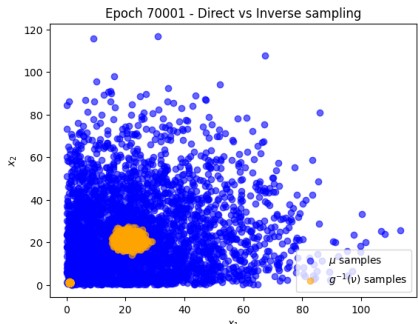

Figure B2: Distributions $\mu$ and $g^{-1}(\nu)$

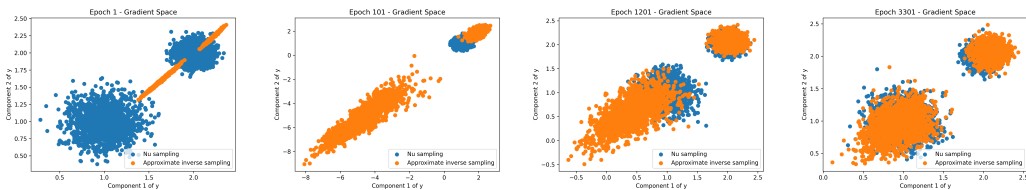

Figure B3: Learning mixture of Gaussian distribution with the inverse sampling: leftmost – train step 1, rightmost – train step 3300.

**Kolmogorov-Arnold neural networks and approximate inverse sampling**

Inspired by the Kolmogorov–Arnold representation theorem, which shows that every continuous function $\phi \colon [0,1]^d \to \mathbb{R}$ can be written as

$$\phi(x) = \sum_{i=1}^{2d+1} \Phi_i \left( \sum_{j=1}^{d} \phi_{i,j}(x_j) \right)$$

for one-dimensional functions $\phi_{i,j} : [0,1] \to \mathbb{R}$ and $\Phi_i : \mathbb{R} \to \mathbb{R}$, a KAN is a composition

$$\boldsymbol{\Phi}_L \circ \ldots \circ \boldsymbol{\Phi}_1 \colon \mathbb{R}^d \to \mathbb{R}^n$$

of KAN-layers of the form

$$\boldsymbol{\Phi}_l(x) = \begin{pmatrix} \sum_{j=1}^{d_{l-1}} \phi_{1,j}^l(x_j) \\ \vdots \\ \sum_{j=1}^{d_{l-1}} \phi_{d_l,j}^l(x_j) \end{pmatrix}, \quad x \in \mathbb{R}^{d_{l-1}},$$

where the one-dimensional functions $\phi_{i,j}^l \colon \mathbb{R} \to \mathbb{R}$ are specified as

$$\sum_i c_i B_i$$

for given basis functions $B_i$ and learnable parameters $c_i \in \mathbb{R}$.

The canonical choice of $B_i$ are polynomial splines with learnable grid-points. But, depending on the situation, some of the $B_i's$ can also be specified as special functions such as $\exp, \log, \sin, \cos, \tan$.

