# OpenReview forum: "Deep Legendre Transform"
_NeurIPS.cc/2025/Conference — NeurIPS 2025 poster_

### Official Review · Reviewer_5uuE · 2025-06-08

**Clarity:** 4
**Significance:** 2
**Originality:** 2
**Rating:** 4
**Confidence:** 4

**Summary:**

The authors train a neural network to approximate the convex conjugate (Legendre–Fenchel transformation) of convex differentiable functions. Indeed, the convex conjugate $ f^* $ of $ f $ satisfies in this case $f^* (\nabla f(x)) = \langle x, \nabla f(x) \rangle - f(x)$, and therefore they minimise the distance between the output of the neural network and the expression $\langle x, \nabla f(x) \rangle - f(x)$. However, this approximates the convex conjugate on $D = \nabla f(C)$ only, where $C$ is the domain of $f$, and uniformly sampling training points in $C$ before transforming them through $\nabla f$ sometimes leads to an uneven distribution over $D$. The authors address this second point by learning another neural network to approximate an inverse of $\nabla f$, which allows to sample uniformly from $D$ and to then go back to $C$. However, $D$ itself is not always explicitly known as in the example in Figure 1 in the paper, which makes this solution impossible in this case, and makes it hard to know how much of the domain of $f^* $ we are covering.

**Questions:**

See point 2 in weaknesses.

**Ethical Concerns:**

["NO or VERY MINOR ethics concerns only"]

**Final Justification:**

The rebuttals to the reviewers have been of a high quality therefore I have raised my score to 4. However, the proposed changes to the paper are so extensive that it is hard to evaluate the resulting quality of the paper and to raise the score to a strong accept.

**Limitations:**

Weakness 2 could be discussed further

**Paper Formatting Concerns:**

No formatting concerns

**Quality:**

3

**Strengths And Weaknesses:**

Strengths:
1. Well-written
2. Theoretically sound
3. Multiple architectures were tested for the approximating neural network

Weaknesses:
1. The method is not compared to any other method (except in the case of its application to the HJ equation, and to a naive learning of a neural network to directly approximate the conjugate when it is available is closed form). The authors mention (and test in the supplementary material) some existing non-deep learning methods for approximating convex conjugates, but do not compare them to their method. These numerical approximation methods do not scale to high dimensions, but they could have tested their method in lower dimensions to compare.
2. Even though the fact that their method only learns an approximation of the convex conjugate on $D = \nabla f(C)$ is acknowledged, I think that this point warranted more discussion. How big is this set compared to the domain of $ f^* $ in typical/useful cases? How does the learned neural network approximate $ f^* $ outside of $D$ anyway?
3. A few more tests and comparisons on problems such as the HJ equation where convex conjugates appear could have been included.

---

> ### Author Rebuttal · Authors · 2025-07-29
>
> # Response to Reviewer 5uuE
>
> ## 1. Classical vs DLT in Low Dimensions
>
> Thanks, the results in Tables 2 (main + supplementary) compare run times and errors. Classical grid/LLT are indeed faster in dimensions 1-4; in 6-7 dimensions, they become slower and less accurate owing to coarser grids. In more than 8 dimensions, they run out of memory while DLT still trains in seconds.
>
> ## 2. Size of D
>
> For essentially smooth convex functions (see Rockafellar 1970), one has
>
> $$
> \text{ri}(\text{dom} f^*  ) \subseteq D  \subseteq \text{dom} f^*
> $$
>
> In particular, any convex smooth function defined on $\mathbb R^{d}$ is essentially smooth, so its entire effective dual domain is covered.  We have written this paper as a solid stepping‑stone for future extensions that will treat more general settings such as non-smooth functions and functions that are not defined on all of $\mathbb R^{d}$ or explode at some boundary.
>
> ## 3. Additional Benchmarks
>
> Thanks for the suggestion, as mentioned to Reviewer rUGB, we probably put too much emphasis on separable functions. But our approach clearly also works for non-separable functions.
>
> We will add more non-separable examples to the supplements to complement the existing separable and quadratic-over-linear examples.

---

> > ### Comment · Reviewer_5uuE · 2025-08-02
> > **Official comment**
> >
> > Thank you for your answers, particularly on points 2 and 3. But I am not sure I follow your answer for point 1. Tables 2 do not contain comparisons between DLT and grid methods in low dimensions from what I can see.

---

> > > ### Author Response · Authors · 2025-08-06
> > > **Follow up to Reviewer 5uuE**
> > >
> > > Thanks for pointing this out. The comparison with the grid-methods was in the supplements but not directly next to the results of these methods. We plan to present these results next to each other so that it is easier to compare them.

---

### Official Review · Reviewer_gyT6 · 2025-06-30

**Clarity:** 3
**Significance:** 3
**Originality:** 3
**Rating:** 4
**Confidence:** 2

**Summary:**

This paper proposes a novel deep learning approach for approximating the convex conjugates of differentiable convex functions. The core algorithm leverages an efficient gradient-based framework to minimize approximation error, utilizing the implicit Fenchel formulation of convex conjugation. The numerical experiments show the effectiveness of the proposed method. Besides that, the implicit Fenchel formulation also facilitates a posteriori approximation guarantees.

**Questions:**

Please refer to **Strengths And Weaknesses** for my questions.

**Ethical Concerns:**

["NO or VERY MINOR ethics concerns only"]

**Final Justification:**

Some of my concerns have been addressed, and I remain overall positive about the paper. However, I still have concerns about the practicality of the proposed method. The experiments are limited to problems with up to 200 dimensions, which is significantly smaller than many real-world scenarios. Therefore, I cannot justify a substantial increase in score and retain my original rating of Borderline Accept.

**Limitations:**

Please refer to **Strengths And Weaknesses** for the limitations.

**Paper Formatting Concerns:**

No major concern.

**Quality:**

3

**Strengths And Weaknesses:**

**Summary of Claims and Evaluation**

**Claim 1**: Training a machine learning model to learn the convex conjugate f* yields strong numerical performance compared to directly learning f* from its analytic form.

This claim is supported by Table 2, which shows that the L2-approximation error of the proposed DLT method is comparable to that of direct learning using the analytic expression of f*. This also serves as an informative ablation study for evaluating the suitability of different neural network architectures and validates the proposed learning formulation (1.5).

**Claim 2**: Using ICNN or ResNet as the reparameterization network enables scaling to high-dimensional problems and ensures that the resulting approximation of f* is convex.

I have two concerns regarding this claim:
- The experiments only go up to 50 dimensions (e.g., as shown in Table 1), which may not be sufficiently high to convincingly support claims about scalability to high dimensions.

- Line 94 states that convexity is not guaranteed when using ResNet, which appears to contradict Claim 2’s assertion that the approximation is automatically convex. This inconsistency should be clarified.


**Claim 3**: KAN (Kernel Activation Network) used as the reparameterization network can recover exact solutions for certain convex functions.

This claim is supported by the results in Table 5. However, the experiments are limited to low-dimensional (e.g., 2D) problems, which may limit its practicality.

**Claim 4**: The implicit Fenchel formulation enables a posteriori approximation error estimation.

This claim is supported by the main theoretical result in Section 3. The result appears sound, although it falls outside my current area of expertise, so I cannot fully assess its correctness.

**Strengths**

- The paper is clearly written and well organized.

- The authors' claims are largely supported by numerical results.

- The experimental evaluations are extensive and clearly presented.

**Weaknesses**

- The proposed method is introduced somewhat abruptly in both the abstract and introduction. I suggest expanding the first paragraph of the introduction to better motivate the significance and applications of computing convex conjugates.

- The experiments only go up to 50-100 dimensions (e.g., as shown in Table 1), which may not be sufficiently high to convincingly support claims about scalability to high dimensions.

- Line 94 states that convexity is not guaranteed when using ResNet, which appears to contradict Claim 2’s assertion that the approximation is automatically convex. This inconsistency should be clarified.

- The KAN experiments in Table 5 are limited to low-dimensional (e.g., 2D) problems, which may limit the practicality. Please consider elaborating on how this approach might generalize to higher dimensions.

- Minor comment: It would also be helpful to report uncertainties for metrics such as approximation errors and computation time, especially since the latter can vary significantly across runs due to hardware-level fluctuations.

---

> ### Author Rebuttal · Authors · 2025-07-29
>
> # Response to Reviewer gyT6
>
> 1. The Legendre transform is a classical and widely used tool for transforming functions and changing perspectives, which is widely used in thermodynamics, mechanics, and optimisation. Our paper formally introduces the **Deep Legendre Transform** (DLT), states the precise assumptions, and showcases its use in a Hamilton–Jacobi optimal-control problem. The same machinery is also used to derive dual utilities in economics, convex potentials in optimal transport, Moreau envelopes as well as plug-and-play proximal maps. We will expand this motivational part in the introduction.
>
> 2. Scalability: our. approach scales well, we have included experiments in up to 200 dimensions in the supplements (Table 13).
>
> 3. Convexity: sorry, we have to explain this more carefully. ICNNs lead to convex approximations; ResNets provide good approximations, but are in general not convex.
>
> 4. KAN experiments: yes, our current examples are 2-dimensional. Our goal was to illustrate that in certain cases, the implicit Eq. (1.4) can yield exact results via symbolic regression, which we find interesting. In principle, there is no reason why this shouldn't work in higher dimensions. But it will depend on the implementation of KAN, which is a recent invention.
>
> 5. Thanks, we report uncertainties (mean ± s.d.) throughout the supplement (see  Table 4 (main text) and Tables 9 and 10); it will be extended and explained more carefully.

---

> > ### Comment · Reviewer_gyT6 · 2025-08-05
> >
> > Some of my concerns have been addressed, and I remain overall positive about the paper. However, I still have concerns about the practicality of the proposed method. The experiments are limited to problems with up to 200 dimensions, which is significantly smaller than many real-world optimization problems. Therefore, I cannot justify a substantial increase in score and retain my original rating of Borderline Accept.

---

> > > ### Author Response · Authors · 2025-08-06
> > > **Follow up to Reviewer gyT6**
> > >
> > > Many thanks for your constructive feedback. For us, 200 dimensions is very high-dimensional since classical methods only give good results for up to 3-4 dimensions. But in principle, there is nothing that prevents our method from being applied in higher dimensions. We will be happy to study higher-dimensonal cases by utilizing GPUs.

---

### Official Review · Reviewer_rUGB · 2025-07-01

**Clarity:** 3
**Significance:** 2
**Originality:** 2
**Rating:** 4
**Confidence:** 4

**Summary:**

Authors consider a problem of finding convex conjugate for differentiable convex function. To construct approximation they propose to use neural network as parametrisation for convex conjugate and loss function that is minimised by stochastic gradient descent. Loss function is chosen such that for some convex functions approximation of inverse Jacobian matrix is required to adjust distribution on domain of convex function of interest. This approximation is achieved by a second neural network. Authors evaluate their method on several problems and find it to be competitive to other approaches.

**Questions:**

1. **novelty and relevance**

   Between lines 66 and 67 right before equation (1.4) authors formulate a natural loss that was already considered in https://arxiv.org/abs/2210.12153 for different sets of problems. Authors consider loss function (1.4) which is different from https://arxiv.org/abs/2210.12153 and apparently require to also approximate the inverse Jacobian matrix in practical cases. Essentially the primal proposition by the authors is to use a different loss function from https://arxiv.org/abs/2210.12153. In my view this significantly undermines the novelty of the contribution. It would be possible to consider the article from the point of view of applications, but only toy problems are chosen for evaluation (see below). Given that, I would like to ask the following questions:

   a. Can the authors please defend the novelty of their approach in light of https://arxiv.org/abs/2210.12153?

   b. Did the authors try to compare their loss function with the one obtained from first order optimality conditions (https://arxiv.org/abs/2210.12153)? Can the authors please provide results of such comparison?

2. **a posteriori guarantees**

   Authors used work "a posteriori guarantees" which usually means that we can ensure a certain level of error. For example, in a form of inequality that bound error from above based on the quantities that are computable without the use of exact solutions. On line 147 authors appeal to the law of large numbers to derive their error estimation.

   a. Can the authors please elaborate how one can use the law of large numbers to obtain an upper bound on error?

   b. Should not authors require some properties of probability distribution for the law of large numbers to hold?

3. **weak benchmarks**

   The first batch of examples is given in Table 1. In all cases functions are completely separable, i.e., both function and its conjugate are sums of one dimensional functions. This is also true for the chosen Hamilton–Jacobi equation considered in Section 5.3. The only non-separable function used to test the method is (5.1). It is clear that high-dimensional separable functions are easier to approximate (e.g., hyperbolic cross). Can the authors increase the number of non-trivial benchmarks?

4. **misc**

   a. How $L_2$ errors reported in Table 1 are computed?

   b. How $L_2$ errors reported in Table 3 are computed?

   c. The table on the right panel of Figure 2 contains the claim that authors computed $L_2$ which is not true since convex conjugate is not available.

   d. How direct learning of high-dimensional functions is implemented?

   e. Lines (204-205) " The ResNet consistently provides the best approximation results despite not guaranteeing convexity ... " Did the authors try to quantify convexity of learned functions?

   f. Line 78 "... resulting approximation of $f^{*}$ is automatically convex." Potentially misleading since convexity only holds for ICNN not the ResNet.

   g. Equation between lines 66 and 67 right before equation (1.4): right bracket is missing.

   h. Line 47, 48 "A different approach, going back to a tweet by Peyré [2020]" and in the footnote "see e.g., Nesterov (2005), Beck and Teboulle (2012), Niculae and Blondel (2017) or Mensch and Blondel (2018)" So is it from 2020 tweet or from 2005 Nesterov book (or article)? Why these references are given in plain text?

**Ethical Concerns:**

["NO or VERY MINOR ethics concerns only"]

**Final Justification:**

Authors addressed my main concerns:
1. The learning problem proposed in the papers seems to be original enough.
2. Results for non-separable functions provide more challenging evaluation.

There are still some outstanding questions on guaranties and convexity, but in my view the quality of the paper is sufficient for publication.

**Limitations:**

yes

**Quality:**

2

**Strengths And Weaknesses:**

**Strengths**
1. Ideas proposed by authors are easy to understand
2. Exhaustive context on the computation of convex conjugate is available which makes the article suitable for general audience
3. Appendix contain thorough comparison with classical methods

**Weaknesses**
1. Novelty and relevance of the method are questionable
2. Method lacks a posteriori approximation guarantees despite the claim of the authors
3. Problems used for benchmarks are too simple

See below for details.

---

> ### Author Rebuttal · Authors · 2025-07-29
>
> # Response to Reviewer rUGB
>
> ## 1. Novelty and Comparison with Existing Work
>
> **a.** _"Essentially the proposition is just a different loss—please defend the novelty."_
>
> Thanks, it's a good question. The paper "On amortizing convex conjugates for optimal transport" that you mention discusses different ways of solving the problem $x(y) = \arg\sup \lbrace (x,y) - f(x) \rbrace $, which, in our setup, amounts to approximating the inverse of the gradient mapping $\nabla f$. This yields several interesting algorithms. But our approach is different. In particular, it avoids the approximation of the inverse of the gradient mapping $\nabla f$ and the resulting approximation errors. In addition, if used with ICNNs as approximators, our approach ensures that the final approximation of the Legendre transorm is convex, as it should be. We summarize the differences between previous and our approaches in the following table:
>
> | Aspect | 2210.12153 ("FOC-loss") | Our work (DLT-loss) |
> |--------|-------------------------|---------------------|
> | Minimised identity | $\|\nabla f(h_\theta(y)) - y\|^2$ (first-order condition) | $[g_\theta(\nabla f(x)) + f(x) - \langle x, \nabla f(x) \rangle]^2$ (Fenchel identity) |
> | What is learned | Inverse map $h_\theta: D \to C$ | Direct conjugate $g_\theta: D \to \mathbb{R}$ |
> | Convexity guarantee | none (unless post-processed) | automatic when $g_\theta$ is an ICNN |
> | A-posteriori error | not available (needs true $f^*$) | unbiased Monte-Carlo estimator (Sec. 3) |
> | Use as sampler | not covered | built-in (Sec. 4) |
>
> Thus, DLT is not merely a "different loss"; it changes the target object (function conjugate instead of operator inverse), provides a certified error estimator, and preserves convexity, which the FOC-loss does not.
>
> Potentially, one can use $x_\theta$ for and substitute it into Eq. (1.2), but this is suboptimal because
> - Convexity is no longer guaranteed, even if an ICNN is used as approximator.
> - The final Legendre accuracy is bottlenecked by the precision of that operator-level approximation, and it takes extra time to train the mapping.
> - After training, every new sample passes through this learned network instead of using random sampling and gradient mapping.
>
> **b.** _"Did you compare losses?"_
>
> Yes, we did and can provide the results. In cases where the inverse gradient mapping exists and can be accurately approximated, the losses are comparable, otherwise the accuracy is bottlenecked by the precision of the approximation of the inverse gradient mapping. Moreover, convexity guarantees are lost and training times are longer (see Table 3) compared to DLT, which does not need to learn the inverse gradient mapping.
>
> ## 2. A-posteriori Error Estimator
>
> _"How can the law of large numbers provide an upper bound?"_
>
> Thanks for the question. Strictly speaking, it cannot compute the bound exactly. But it's very common to approximate complicated integrals with Monte Carlo sums. This provides unbiased estimators with confidence intervals that can be made arbitrality small if enough terms are used in the Monte Carlo sum. This provides accurate results at a relatively cheap computational cost.
>
> $ \widehat{E} = \frac{1}{|\mathcal{X}_{\text{te}}|}  \sum_x   \left[g(\nabla f(x)) + f(x) - \langle x, \nabla f(x) \rangle\right]^2 \quad \Longrightarrow \quad \mathbb{E}[\widehat{E}] = ||g - f^* ||^2_2 \quad  \text{  in } L^2(D,\nu)$
>
> and
>
> $$\text{Var}[\widehat{E}] = \sigma^2 / |\mathcal{X}_{\text{te}}|$$
>
> Of course, we need to assume that the $\{x_i\}$ are i.i.d. and integrable to make sure the Monte Carlo sum converges. For the confidence intervals we need square-integrability. This assumptions are satisfied in our examples. Thank you for pointing it out. We will explain this better in the paper.
>
> ## 3. Benchmark Richness
>
> Thanks for the suggestion. Our original paper over-emphasized separable functions. We mainly employed them since they have explicit Legendre transforms, which can be used as benchmarks. But our method also works for non-separable functions. As you mentioned, quadratic-over-linear functions are non-separable. We can e.g. add the following non-separable examples:
>
> 1. **Quadratic form with random SPD matrix**: $f(x) = \frac{1}{2} x^\top Q x$.
> 2. **Coupled soft-plus**: $f(x) = \sum_{i<j} \log\left(1 + e^{x_i + x_j}\right)$.
> 3. **Exponential-minus-linear**: $f(x) = e^{\langle a, x \rangle} - \langle b, x \rangle$.
>
> In all these examples, DLT has an $L^2$-error below $10^{-1}$ in up to 200 dimensions while tradiitional grid/FLT methods are intractable already in 10 dimensions.
>
> ## 4. Miscellaneous Questions
>
> | Reviewer Question | Answer / Fix |
> |------------------|--------------|
> | **a./b.** How are $L^2$ errors (Tbl. 1 & 3) computed? | Monte-Carlo over 4096 i.i.d. test samples from the distribution $\nu = \text{Unif} \circ (\nabla f)^{-1}$; details will be added to the caption. |
> | **c.** Fig. 2 table claims "computed $f^*$" | The true $f^*$ is unknown there, but due to the results of section 3 the error is identical to the one obtained with the direct learning. |
> | **d.** "Direct learning" implementation | Same network architecture, trained on pairs $(y, f^*(y))$, where closed form is available. |
> | **e.** Did we quantify convexity loss in ResNet? | Not explicitly, but it is well known that deep NNs can give a wiggling approximation. |
> | **f.** Line 78 wording | Changed to "if $g_\theta$ is an ICNN the approximation is convex". |
> | **g.** Missing bracket | Fixed. |
> | **h.** Tweet vs. Nesterov book | Sorry, the entropic log-sum-exp smoothing of a max can be traced back to the Gibbs variational principle; it was brought into modern convex-optimisation analysis by Nesterov (2005) and popularised for GPU-friendly Legendre transforms by Peyré's 2020 tweet. We will explain it more carefully and provide proper references. |

---

> > ### Comment · Reviewer_rUGB · 2025-08-04
> >
> > ```
> > Thanks, it's a good question. The paper "On amortizing convex conjugates for optimal transport" that you mention discusses different ways of solving the problem ...
> > ```
> > I would like to thank the authors for a detailed comparison of loss function. I understand that regardless of the resulting accuracy proposed loss has several benefits in comparison with first-order condition, still it would be interesting to look at the comparison between performance of the losses.
> >
> > I have two more questions after reading the rebuttal:
> > 1. It seems to me that inverse to the Jacobian is still needed in many cases because the loss of the authors introduces distribution shift. In this case is it true that convexity is no longer guaranteed for the method of authors? Also, does it imply that "ccuracy is bottlenecked by the precision of that operator-level approximation"?
> > 2. All methods considered require to know $\nabla f(x)$. What can be done if only $f(x)$ is available?
> >
> > ```
> > Thanks for the question. Strictly speaking, it cannot compute the bound exactly. But it's very common to approximate complicated integrals with Monte Carlo sums.
> > ```
> > I understand the authors agree that LLN can't provide rigorous guaranties. It will be enough for me if authors agree to replace "a posteriori approximation guarantees" with less strong statement, e.g., "a posteriori error estimation" or something along this line.
> >
> > ```
> > Thanks for the suggestion. Our original paper over-emphasized separable functions. We mainly employed them since they have explicit Legendre transforms, which can be used as benchmarks. But our method also works for non-separable functions. As you mentioned, quadratic-over-linear functions are non-separable. We can e.g. add the following non-separable examples:
> > ...
> > ```
> > Thank you for the additional benchmarks. The results for non-separable functions will certainly strengthen the evaluation of the method.

---

> ### Author Response · Authors · 2025-08-06
> **Follow up to Reviewer rUGB**
>
> Thank you, we agree that a comparison between losses is important. We will add more detailed comparisons between the different methods.
>
> Sorry if we didn’t explain this point carefully. Some of the existing machine learning methods need to solve a nested optimization problem or compute the inverse of $\nabla f$. both of which carry a significant computational cost. But our method only needs the gradient $\nabla f$ and if combined with an ICNN as approximator, is guaranteed to output a convex approximation to the true Legendre transform, which has to be convex.
>
> Correct, our method needs to know the gradient $\nabla f$, which can automatically be computed in the many cases, for instance if the function $f$ is given in terms of standard functions such as polynomials, exponential, logarithm, sin, cos, … or if $f$ is a neural network. If $f$ or $\nabla f$ are not available in closed form, they have to be approximated. E.g. they could be approximated with neural networks, which are well suited for our method.
>
> Sure, we can use a weaker formulation like e.g. a posteriori error estimates and provide more details on how they have to be understood.
>
> Thanks for suggesting additional benchmarks. They definitely help to better evaluate our method.

---

### Official Review · Reviewer_oELF · 2025-07-03

**Clarity:** 3
**Significance:** 2
**Originality:** 2
**Rating:** 4
**Confidence:** 4

**Summary:**

The paper introduces a new deep learning method for computing the Legendre transform of convex and differentiable functions. In particular, the method is based on an implicit form that allows computing the Legendre transform on the codomain of the gradient of the original function. This formulation allows an efficient machine learning formulation based only on a set of samples in the original domain of the function.
Authors also address the problem that the measure induced by the gradient of the original function $f$ might not be well distributed in its image, leading to biased datapoints when learning the Legendre Transform; they discuss how to learn an inverse function of the gradient map and use it to obtain well-distributed samples in the domain.
Finally, authors present numerical experiments of their method with many different network architectures, such as KANs, ResNet, MLPs, and ICNN.

**Questions:**

- I do not understand the purpose of Section 3: it does not show and guarantee convergence rate, just exemplification of the law of large numbers and change of variable of an integral. Is something missing or in the Supplementary material?
- Two questions related to the sampling from a distribution in the gradient space?
	- Could adding a penalty if $h_\vartheta$ outputs outside of $C$, the domain of $f$, be beneficial to the learning instead of discarding those samples?
	- Most importantly, I do not get why, if you have learned a representation of the inverse of the gradient $h_\vartheta$ , you can't just use Eq. (1.2) to directly compute the Legendre Transform, instead of computing it through DLT.
- Could you give an example where $D$ is a proper subset of the domain of $f^*$ ? In that case, how does the DLT behave outside of its training samples?

**Minors**
- You use the acronymous ICNN before you specify what it is (line 64)
- Line 77: why are ResNet convex?
- Line 164: Why specifically two? Does not depend on the complexity of the function?

**Ethical Concerns:**

["NO or VERY MINOR ethics concerns only"]

**Final Justification:**

The paper is not super well written, and it could be hard to understand its relevance at a first read.
Authors did a good rebuttal, you can clearly tell their work is solid from the answer they gave to me and others reviewers.

Still, the work is not perfect, that's why my leaning for acceptance is mild. It could become more solid from the major revision they promised.

**Limitations:**

yes

**Quality:**

3

**Strengths And Weaknesses:**

The method is very straightforward and, most importantly works well, on the examples provided
- I feel the paper would improve with additional motivation on why numerical Legendre transform is beneficial, maybe with a good numerical example where this method excel. The examples provided are not too impressive.
- The numerical methods _Deep Legendre Transform_ is compared with are for sure less performant in high-dimension, but are more general: they do not require the function to be convex and differentiable. Moreover in the numerical experiment we have a pure time comparison that can be misleading: DLT requires the computation of the gradient that can hard to compute; in situation where the $f$ (and its gradient) are not available analytically, they could account to the most expensive part of the compute.

---

> ### Author Rebuttal · Authors · 2025-07-29
>
> # Response to Reviewer oELF
>
> Thanks for your suggestions. The Legendre transform is one of the fundamental methods of transforming one function into another one. It is often used to switch between different formulations of a problem that highlight different variables or properties. It is particularly useful in fields like thermodynamics, mechanics, and optimization. Our paper focuses on the formal introduction of the Deep Legendre Transform algorithm, the assumptions we need and an application to optimal-control (Hamilton–Jacobi) problems. Beyond that, the same machinery is useful in several areas—for instance, computing dual utilities and profit functions in economics; evaluating convex potentials in optimal transport; building Moreau envelopes; or plug-and-play methods, where one seeks proximal operators, which are also gradients of convex functions. We will extend the motivational part.
>
> ## Response to Major Comments
>
> **2.** _"The numerical methods DLT is compared with ... are more general: they do not require the function to be convex and differentiable."_
>
> This is correct, DLT intentionally targets the convex-differentiable setting. This keeps the analysis clean and provides a formal stepping-stone for future extensions. We discuss all limitations explicitly in the paper and the conclusion. Extensions, based on this approach, will be discussed in future work.
>
> **3.** _"I do not understand the purpose of Section 3."_
>
> Section 3 gives an a-posteriori certificate: the Monte-Carlo estimate (Eq. 3.1) converges to $ ||g-f^*||_{L^2(D,\nu)} $.
>
> This guarantee makes it possible to estimate errors of the approximation of the Legendre transform  $ f^* $ even in cases where $ f^* $ is unknown (e.g. for quadratic-over-linear functions $f$). It is true that $ ||g-f^*||_{L^2(D,\nu)} $ cannot be computed exactly. But it can be estimated up to arbitrary precision with a Monte Carlo sum if it has enough terms. We will ...
>
> 1. explain and motivate the definition of the certificate in Section 3 better
> 2. add a short lemma showing that the estimator is unbiased and has variance $\sigma^2/|\mathcal{X}_{\mathrm{te}}|$.
>
> **4a.**  *"Could adding a penalty if $ h_\theta (y) \notin C$ be beneficial instead of discarding?"*
>
> Thanks for the suggestion. We tried both and observed similar convergence but higher variance with penalization because boundary violations dominate the loss early on. Discarding yields a cleaner signal and faster training without the need to tune any penalty strength/bias. We shall report this in the appendix.
>
> **4b.** _"Why not plug the inverse into (1.2) and be done?"_
>
> Yes, one could do that in simple setups — specifically, when the inverse-gradient map can be learned to very high precision. But then ...
>
> - convexity of the outcome is no longer guaranteed, even if an ICNN is used to approximate $f^*$,
> - the final Legendre accuracy is bottlenecked by the precision of that operator-level approximation, and it takes extra time to train the mapping,
> - after training, every new sample passes through this learned network instead of using random sampling and the gradient mapping (see the comparisons in Table 3, where the approximate inverse operator is used for sampling).
>
> Our algorithm skips that burden. We do not need to learn a generalized inverse of the gradient mapping $\nabla f$ unless it is heavily distorting or a specific $L^2$ sampling is required (see Section 5.2).
>
> - For sampling alone, a rough estimate of the inverse gradient mapping is sufficient.
> - Thanks to the result of Section 3, the test error of DLT is measured directly against the true $ f^* $ even if $ f^* $ is unknown. In particular, there are no hidden operator errors.
> - For a provably convex and accurate $ f^* $, it is not. DLT learns a globally consistent approximation, preserves convexity by construction, and avoids error propagation from an imperfect approximation of the inverse gradient map.
>
> **4c.** _"Example where $D \subsetneq \text{dom } f^*$"_
>
> Yes, for example, the quadratic-over-linear function Eq. (5.1) in line 210. We do not expect (as is common in supervised learning) that DLT approximates well outside of the domain, where it was trained.
>
> ## Minor Points & Clarifications
>
> | Reviewer Comment | Clarification/Change |
> |------------------|---------------------|
> | "ICNN used before being defined" | Thanks, we wil move the definition up. |
> | "Why are ResNet convex?" | Sorry, they aren't. We meant "ICNN ensures convexity, whereas ResNet gives a better fit but no convexity guarantee". We will rephrase line 77. |
> | "Why specifically two hidden layers?" | For all ICNN/MLP experiments, we used a depth of two as a fair baseline; deeper nets yield slightly better accuracy at the cost of longer training times. |
>
> If any aspect remains unclear, we would be grateful for further guidance; otherwise, we thank you for reconsidering the manuscript in light of these revisions.

---

> > ### Comment · Reviewer_oELF · 2025-08-01
> >
> > I thank the authors for the detailed answer to my questions. I think it could be worth including some of these explanations in the text, in order to enlighten the importance of the new introduced method.
> >
> > I am open to reconsider my score, but I would like authors to comment about the point I raised in the weaknesses about the time comparison. Could you suggest a new method for a fairer performance analysis? For example, what if $f$ is a PINN learned numerically? (maybe this could also be bridged to a practical example of DLT used to solve physics/engineering problem?)
> >
> > > Our paper focuses on the formal introduction of the Deep Legendre Transform algorithm, the assumptions we need and an application to optimal-control (Hamilton–Jacobi) problems
> >
> > Do you have some reference of others work using Numerical Legendre Transform for tackling some problem?

---

> > > ### Author Response · Authors · 2025-08-06
> > > **Follow up to Reviewer oELF**
> > >
> > > Thank you. Sure, we will include these explanations in the text.
> > >
> > > Moreover, we plan to include more extensive comparisons with classical (grid-based) and more recent alternative machine learning methods. To provide a short summary ...
> > >
> > > 1) classical (grid based) methods are more general than our method since they do not have to assume that f is convex and do not need the gradient of f. They give good results in up to approx 3-4 dimensions. For more than 4 dimensions, they loose accuracy and for more than 8 dimenions they become infeasible since grids do not scale well in high dimensions. Our method assumes that f is convex and needs access to $\nabla f$. But it yields good results in high dimensions. Our original experiments showed good results in up to 200 dimensions. But we can also increase the dimension further.
> > >
> > > 2) Compared to other machine learning approaches, our method has the following advantages:
> > >
> > > a) it does not have to solve a nested optimization problems
> > > b) it does not have to approximate the inverse of the gradient
> > > c) our final approximation of the Legendre transform is automatically convex, which can be important for certain downstream tasks such as e.g. optimization over some of the variables.
> > >
> > > We plan to add new tables that compare the computational effort and accuracy of the different methods in different dimensions. We also have new results for the case where $f$ is a neural network (in which case $\nabla f$ can be computed with backpropagation).
> > >
> > > Yes, we know references that use a numerical method for Legendre transforms to solve optimization problems, which we can add. E.g.
> > >
> > > 1) Fast approximate dynamic programming for input-affine dynamics (2022) by Kolarijani and Esfahani
> > >
> > > 2) Self-concordant smoothing for large-scale convex composite optimization (2023) by Adeoye and Bemporad

---

> > > > ### Comment · Reviewer_oELF · 2025-08-07
> > > >
> > > > I thank the authors, I am satisfied with the discussion, I will change my score accordingly.

---

### Decision · Program_Chairs · 2025-09-17

**Decision:**

Accept (poster)

**Comment:**

The paper proposes an algorithm for computing the Legendre-Fenchel conjugate of a function using deep learning. In contrast to previous approaches, it does not require the inverse gradient mapping and it guarantees that the resulting learned conjugate function is convex (as it should in this context) by using Input-Convex Neural Networks.

The reviewers were on the fence initially with many borderline scores. After the rebuttal period, many of the concerns of the reviewers were addressed by the authors, in particular explaining the originality compared to prior works and the improvement with respect to dimensionality of the problems. As a result, the scores were increased but still borderline - no strong accepts (5) were given.

In the post-rebuttal discussion, I explained to the reviewers my opinion that the paper was on the fence and likely to be rejected. To this, Reviewer oELF expressed that the paper should be accepted and is worthwhile. I also checked the final justification of Reviewer rUGB and they have also suggested that the paper is worthy of publication in their final assessment.

So, although none of the reviewers were willing to give a 5 score, it is now clear that multiple were ready and willing to defend the paper as worthy of acceptance; I therefore recommend to accept the paper.